# Fidelity in co-diversified symbiosis

Inès Pons ●[1]✉, Marleny García-Lozano ●[1], Christiane Emmerich[1], Aftab Mahmood Ayas ●[1], Christine Henzler[1], Hagay Enav[2], Ruth E. Ley ●[2] & Hassan Salem ●[1,3]✉

Obligate co-dependence can arise in symbiosis, yielding heritable partnerships. These interactions are considered to be highly specific, but partner fidelity is difficult to quantify owing to the experimental constraints of symbiont exchange between host species. Here, we overcome this challenge by leveraging the unique transmission dynamics of *Stammera capleta*, the obligate digestive symbiont of tortoise beetles (Chrysomelidae: Cassidinae). Despite its extracellular localization, *S. capleta* possesses a drastically reduced genome ( ~ 0.25 Mb) and is vertically transmitted through egg-associated spheres. Manipulating these spheres allowed us to experimentally exchange *S. capleta* between beetle species to determine their impact on host development. We show that non-native *S. capleta* can successfully colonize the symbiotic organs of a novel host, but that the interaction outcome correlates with genetic relatedness to the native symbiont. Genetically distant species trigger a more pronounced transcriptional response and can only partially rescue host development. While more closely related symbionts proliferate similarly to the native one and induce a comparable host response, they fail to propagate to the next generation, underscoring how transmission fidelity, host-symbiont compatibility, and local adaptation can further specificity within a Paleocene-aged partnership.

The metabolic intimacy of symbiosis demands the work of specialists, each evolving distinct yet complementary roles to sustain the partnership[1,2]. Obligate co-dependence can arise within these interactions[3,4], necessitating adaptations that ensure the recognition, regulation, and persistence of mutualisms[5–7]. The resulting partnerships often feature heritable endosymbionts that are intertwined in the evolutionary history of their hosts[8–10]. As symbiont and host co-diversify, they exhibit parallel lineage splitting over evolutionary timescales[11].

Among animals, obligately beneficial symbionts are typically acquired early in development[2,4,12–14]. For insects that host endosymbionts within specialized cells (i.e., bacteriocytes) and organs, microbial colonization can take place during embryogenesis or oogenesis[13–15]. For example, aphids acquire their nutritional endosymbiont, *Buchnera aphidicola*, during embryo development through

calibrated cycles of exocytosis and endocytosis[13]. Endosymbiont cells released from maternal symbiotic organs colonize cells fated to become bacteriocytes in the embryo[13]. Ants similarly integrate their endosymbiont, *Blochmannia floridanus*, during embryogenesis through a shared regulatory network that is coopted by the microbe to facilitate infection[14]. Such models, and others[6,15–18], continue to offer striking insights into the cellular mechanisms guiding symbiont colonization and proliferation[12]. But our inability to cultivate most obligate symbionts in pure culture[18], compounded by the brief developmental window that their hosts spend aposymbiotically[6,13,14], constrains the experimental exchange of symbionts and the ability to test fidelity within co-diversified partnerships.

In contrast to the experimental challenges posed by heritable endosymbionts, partnerships with beneficial microbes acquired from the environment have provided crucial insights into the mechanisms

[1]Mutualisms Research Group, Max Planck Institute for Biology, Tübingen, Germany. [2]Department of Microbiome Science, Max Planck Institute for Biology, Tübingen, Germany. [3]Department of Molecular Microbiology, John Innes Centre, Norwich, UK. ✉e-mail: ines.pons-guillouard@tuebingen.mpg.de; hassan.salem@jic.ac.uk

that govern specificity in animal-microbe symbioses[19-25]. For example, the Hawaiian bobtail squid (*Euprymna scolopes*) can differentiate its bioluminescent symbiont *Vibrio fischeri* from thousands of marine microbial species through an intricately honed process that relies on mechanical, chemical, immunological, and developmental exclusion[26-29]. Similarly, the bean bug (*Riptortus pedestris*) acquires its crypt-associated gut symbiont *Caballeronia insecticola* every generation from the soil[19,30-32]. Despite a horizontal acquisition mode[30], several factors contribute to a highly selective recognition process, including a microbial-sorting structure within the gut[20], the competitive exclusion of non-symbiotic strains[33], and, finally, the morphological differentiation and closure of the symbiotic organ following passage by *C. insecticola*[31]. While horizontally transmitted symbionts lack evolutionary histories that mirror their hosts[19,22], the partnerships between the squid-*V. fischeri* and bean bug-*C. insecticola* display striking adaptations for partner recognition at the genetic and molecular levels[20,21,31,33-35]. Whether these features extend to heritable endosymbionts, however, remains largely unexplored across study systems.

Here, we investigated the specificity of a co-diversified symbiosis by examining the partnership between tortoise leaf beetles (Chrysomelidae: Cassidinae) and their digestive bacterial symbiont, *Candidatus* Stammera capleta (Fig. 1A–H)[36-38]. The tortoise beetle-*S. capleta* partnership offers a tractable experimental framework to test fidelity in an obligate symbiosis given the microbe's unique transmission route and colonization dynamics[36,39,40].

Tortoise beetles maintain *S. capleta* in foregut symbiotic organs to facilitate folivory and in ovary-associated glands to ensure transmission (Fig. 1A–D)[36-38,40-42]. Despite possessing a drastically reduced genome (~0.2–0.3 Mb), *S. capleta* encodes and exports several plant cell wall-degrading enzymes, including polygalacturonase, a pectinase that underpins convergent nutritional symbioses across leaf beetles[36-38,43-45]. Tortoise beetles vertically transmit *S. capleta* by depositing an individual caplet at the anterior pole of each egg[36,39] (Fig. 1E). This transmission mode is reflected in the strict co-cladogenesis between symbiont and host[37,38,41]. The caplet is populated with symbiont-bearing spheres where the microbe is embedded extracellularly (Fig. 1F–H)[36,39]. Embryos acquire *S. capleta* late in development by piercing the caplet membrane and consuming the spheres[39], which initiates infection[40].

Removing the egg caplet disrupts symbiont transmission, yielding *S. capleta*-free (aposymbiotic) larvae that exhibit a diminished digestive phenotype and drastically reduced survivorship[36,39,40]. However, reapplying the spheres to aposymbiotic eggs experimentally reconstitutes the symbiosis[39,40], highlighting a potential experimental mechanism to exchange the microbe across different species of tortoise beetles by leveraging these structures.

In this study, we capitalized on the conserved symbiont transmission route shared by tortoise beetles to demonstrate that: (i) heritable endosymbionts can be exchanged between host species, (ii) non-native symbionts can colonize and differentially restore survivorship in a novel host, with outcomes ranging from full fitness recovery to reduced survival depending on genetic divergence among symbionts, (iii) but that, ultimately, a high level of fidelity governs this partnership during host development and the symbiont's propagation to the next generation. Our findings highlight the key, complementary roles that partner recognition, transmission fidelity, and local adaptation all play to ensure specificity and stabilize an ancient, co-diversified symbiosis.

## Results and discussion

### Reciprocal symbiont exchange in tortoise beetles
Symbiont acquisition in tortoise beetles is governed by a strict developmental window[39,40]. Although the foregut symbiotic organs form 3 days before larval emergence from the egg, we previously identified

the final 24 h of embryogenesis as the critical period for *S. capleta* colonization[39,40]. Reapplying symbiont-bearing spheres during this period can restore *S. capleta* infection in aposymbiotic embryos[39,40]. Given this, we explored whether the same approach could facilitate the exchange of *S. capleta* between different beetle species.

Using the tortoise beetle *Chelymorpha alternans* (Fig. 1A) as a model, we validated this method through four experimental treatments: (a) untreated control, (b) eggs with caplets removed (aposymbiotic), (c) eggs with caplets removed but re-supplied with their original symbiont-bearing spheres at the anterior pole (re-infected), and (d) eggs with caplets removed but instead supplied with spheres collected from *Chelymorpha gressoria*, a closely related species (cross-infected) (Fig. 2A).

Consistent with previous studies[36,39,40], caplet removal disrupted *S. capleta* transmission, yielding aposymbiotic embryos, in contrast to the untreated and re-infected controls (Figs. 2A and S1i). Notably, cross-infected embryos (i.e., supplied with spheres from *C. gressoria*) were successfully colonized by non-native *S. capleta* (Figs. 2A and S1i). The symbiont occupied the foregut symbiotic organs of its new host, in line with the colonization dynamics observed for the native *S. capleta* of *C. alternans* (Figs. 2A and S1i).

To determine whether symbiont exchange is reciprocal across tortoise beetle species, we repeated these assays using *C. gressoria* as a host (Fig. 2B). The same four experimental treatments were applied: (a) untreated control, (b) aposymbiotic, (c) re-infected, and (d) cross-infected (i.e., aposymbiotic *C. gressoria* supplied with spheres collected from *C. alternans*). The colonization dynamics mirrored the findings observed in *C. alternans* (Figs. 2B and S1ii). Aposymbiotic *C. gressoria* lacked their native *S. capleta*, in contrast to the untreated and re-infected controls. On the other hand, cross-infected embryos were successfully colonized by the symbiont of *C. alternans* (Figs. 2B and S1ii), indicating that the cross-infection protocol was reciprocal.

To confirm whether these colonization dynamics persist beyond the embryo stage, we evaluated the localization of each symbiont in 5-day-old larvae of both *C. alternans* and *C. gressoria* (Fig. 2A, B). The same patterns were observed across both beetle species: aposymbiotic insects remained uncolonized, while untreated and re-infected controls maintained the native *S. capleta* of each species (Figs. 2A, B and S1iii, iv). Finally, non-native symbionts continued to occupy the foregut symbiotic organs of their new hosts (Figs. 2A, B and S1iii, iv). These findings demonstrate that obligate, heritable symbionts can be experimentally exchanged between tortoise beetle species, and that reciprocal cross-infection persists beyond larval eclosion. With this established, we next asked: how do non-native symbionts influence the development of their novel hosts?

### Non-native symbionts spur a gradient of mutualistic outcomes
Numerous insect traits are endowed through symbiosis[46], with obligate dependence shaping interactions across diverse clades such as bugs[47-49], ants[14], termites[50,51], bees[52,53], and beetles[5,6,54]. Insects specializing on nutritionally imbalanced diets continue to offer important insights into these obligate interactions and the extent to which host development is stunted in their absence[55,56]. For example, aposymbiotic aphids deprived of *Buchnera*—and the essential amino acids it supplements — suffer significantly reduced survivorship and complete loss of fecundity[57,58]. Similarly, bedbugs, unable to balance the nutritional deficiencies of their bloodmeal, are developmentally constrained in the absence of their B vitamin-supplementing endosymbiont[59]. Such findings are consistent with consequences of aposymbiosis in tortoise beetles, where the experimental loss of *S. capleta* impairs the insects' ability to process a leafy diet, resulting in low survivorship and failure to reach adulthood[36,39]. Given these pronounced effects, we asked whether non-native *S. capleta* could rescue aposymbiotic mortality. And if so, how do these beetles develop compared to larvae colonized by the native symbiont?

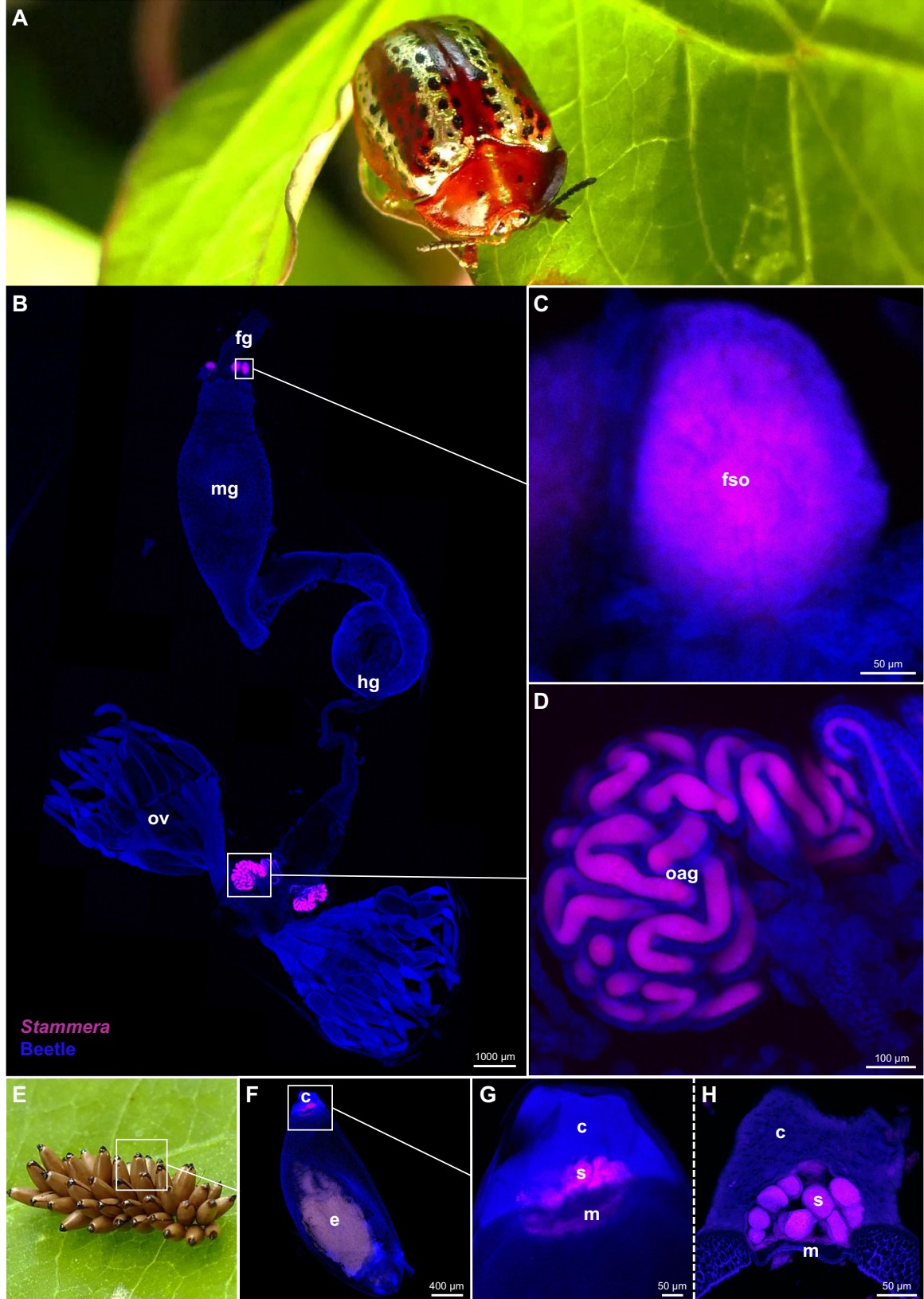

**Fig. 1 | Localization and transmission of *Stammera capleta* in tortoise beetles.**
**A** Adult tortoise beetle (Coleoptera: Cassidinae: *Chelymorpha alternans*). **B** Whole-mount fluorescence in situ hybridization (FISH) demonstrating *S. capleta* localization within each type of symbiotic organ. **C** Foregut symbiotic organs. **D** Ovary-associated glands. **E** Egg mass bearing caplets deposited at the anterior pole of each egg. **F**, **G** Whole-mount FISH of an egg and its caplet. **H** FISH cross-section of a caplet showing enclosed spheres, where *S. capleta* is embedded. Each image is from a single individual and is shown for illustrative purposes. Probes used: *S. capleta* (magenta: 16S rRNA) and the beetle host (blue: 18S rRNA). fg foregut, fso foregut symbiotic organs, mg midgut, hg hindgut, ov ovaries, oag ovary-associated glands, e embryo, c caplet, s *S. capleta*-bearing spheres, m caplet membrane. Scale bars are included for reference.

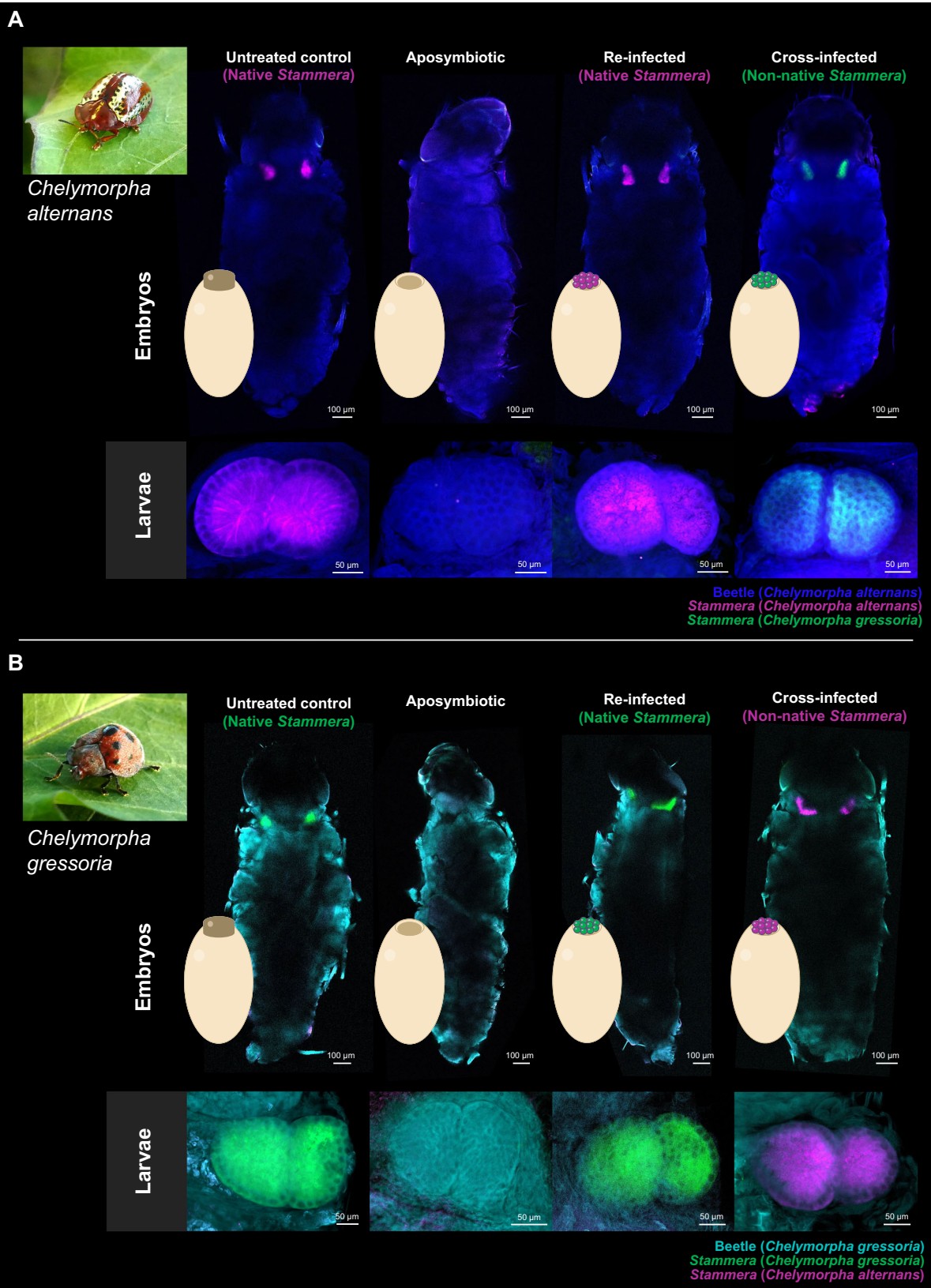

**Fig. 2 | Experimental exchange of *Stammera capleta* in tortoise beetles.** Fluorescence in situ hybridization (FISH) whole-mount images of entire embryos and foregut symbiotic organs in larvae of **A** *Chelymorpha alternans* and **B** *Chelymorpha gressoria* across four experimental treatments: untreated control, aposymbiotic (caplets removed), re-infected with the native symbiont, and cross-infected with the non-native symbiont. Probes used: *S. capleta* from *C. alternans* (magenta: 16S rRNA), *S. capleta* from *C. gressoria* (green: 16S rRNA), *C. alternans* host (blue: 18S rRNA), and *C. gressoria* host (cyan: 18S rRNA). Symbiont-specific probes were applied simultaneously during hybridization. Each image is representative of one individual; similar results were observed in all replicates, with most treatments having three replicates (see Fig. S1i-v). Scale bars are included for reference.

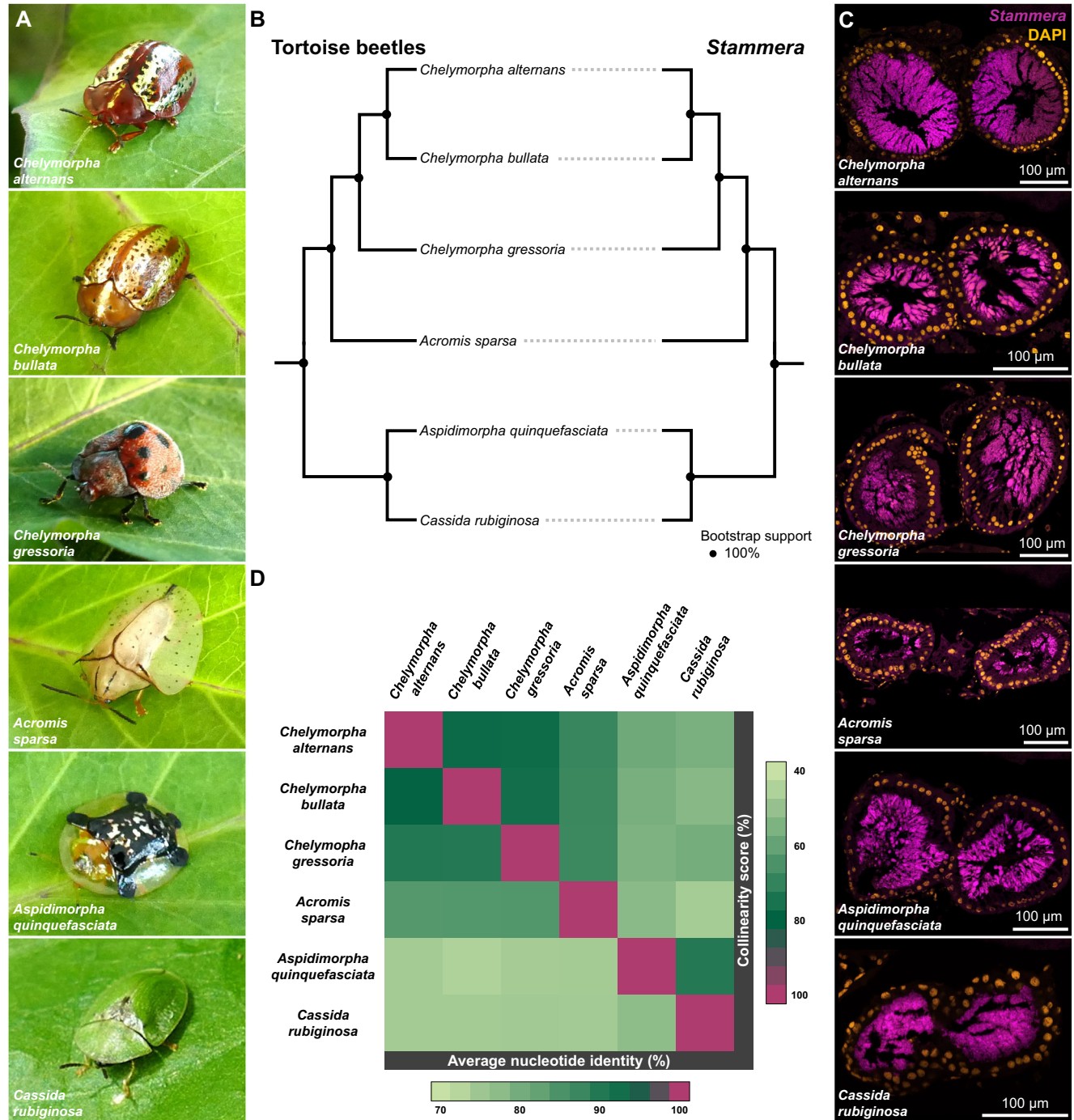

**Fig. 3 | Tortoise beetles co-diversifying with *Stammera capleta*. A** Tortoise beetles, top to bottom: *Chelymorpha alternans, Chelymorpha bullata, Chelymorpha gressoria, Acromis sparsa, Aspidimorpha quinquefasciata,* and *Cassida rubiginosa*. **B** Co-cladogenesis between tortoise beetles and their *S. capleta* symbionts based on Maximum Likelihood (ML) phylogenies (Test of Co-cladogenesis, *p* = 0.0099). The host tree leveraged a concatenated alignment of 15 mitochondrial genes, while the *S. capleta* phylogeny is based on a concatenated alignment of 61 single-copy core genes. Node indicates bootstrap support. **C** Fluorescence in situ hybridization (FISH) cross-sections of foregut symbiotic organs from the beetle species listed above targeting their *S. capleta* symbionts (magenta: 16S rRNA) and DAPI-stained DNA (orange). Each image is from a single individual and is shown for illustrative purposes. Scale bars are included for reference. **D** Heatmap illustrating genetic distance between different *S. capleta* symbionts. Pairwise comparisons of average nucleotide identity (ANI) (light green: 70%; dark green: 95%) and collinearity scores (light green: 40%; dark green: 85%) are shown. Source data are provided as a Source Data file.

We pursued the above assays using six species of tortoise beetles: the aforementioned *C. alternans* and *C. gressoria*, and in addition, *Chelymorpha bullata, Acromis sparsa, Aspidimorpha quinquefasciata,* and *Cassida rubiginosa* (Fig. 3A, and Supplementary Data 1). Each of these beetles species harbors a single, genetically distinct species of *S.*

*capleta* within its foregut symbiotic organs[36–38] (Figs. 3B–D and S2–3), which is transmitted extracellularly via egg-associated spheres[60].

Using *C. alternans* as a host, we assessed the colonization efficiency of different *S. capleta* species in a novel host and quantified their effects on larval survivorship. We structured our bioassays

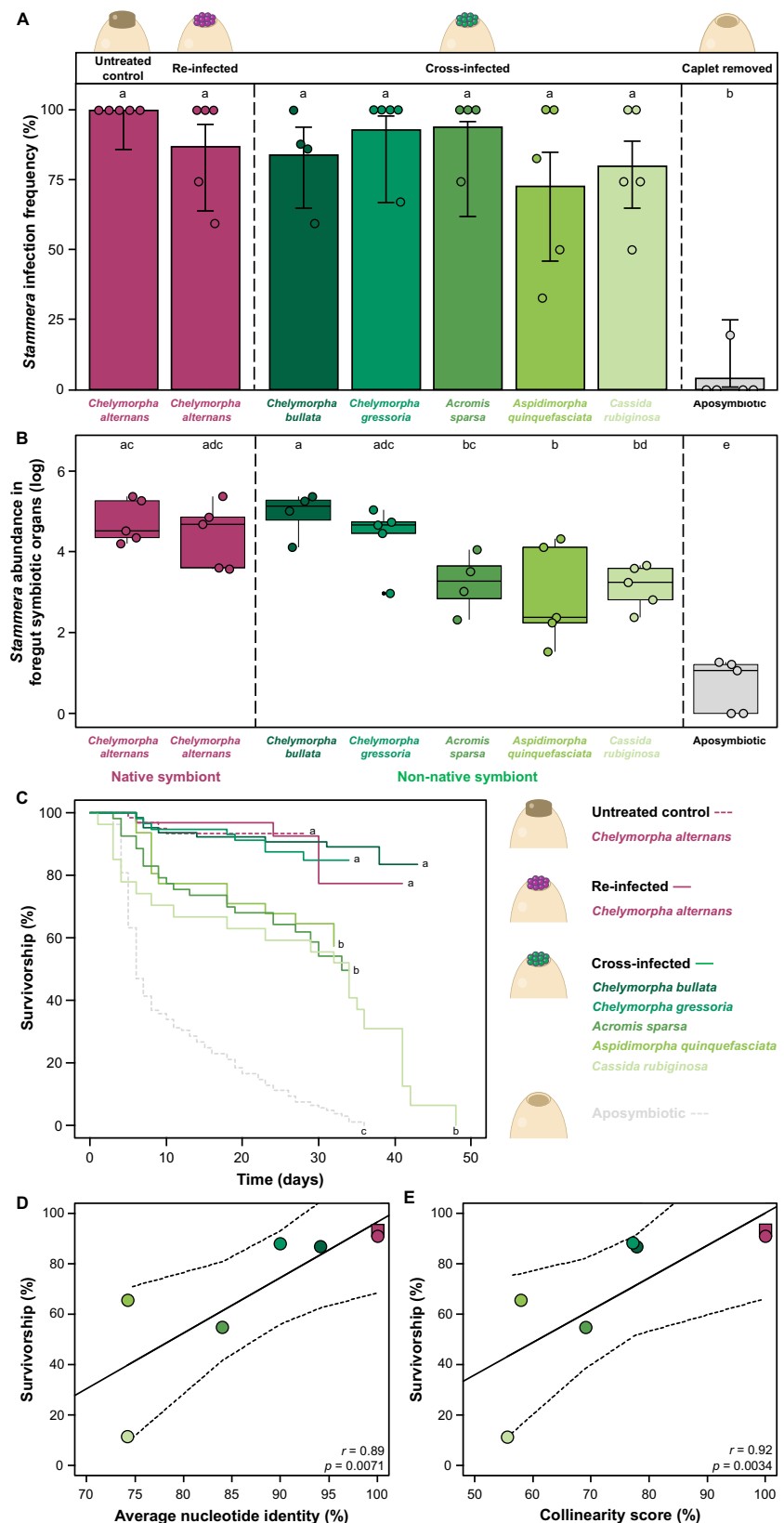

around eight experimental treatments: (a) untreated control, (b) aposymbiotic eggs, (c) eggs whose caplets were removed but re-infected with the native symbiont, or (d–h) eggs whose caplets were removed but cross-infected with *S. capleta*-bearing spheres collected from each of *C. bullata*, *C. gressoria*, *A. sparsa*, *A. quinquefasciata*, and *C. rubiginosa*.

Five-day-old larvae were dissected to assess their symbiotic status. Experimental manipulation significantly influenced *S. capleta* colonization (Fig. 4A; Fisher's Exact Test, $p = 0.0005$; Supplementary Data 2A). As demonstrated earlier (Fig. 2), removal of the egg caplet disrupted symbiont transmission relative to the untreated and re-infected controls (Fig. 4A; pairwise comparisons of proportions,

**Fig. 4 | Non-native symbionts differ in their mutualistic potential. A** Symbiont infection frequency in the foregut symbiotic organs of 5-day-old *Chelymorpha alternans* larvae. Bar colors represent experimental treatments: untreated control (5 rep ($n = 24$)), aposymbiotic (5 rep ($n = 19$)), re-infected with the native bacterium (5 rep ($n = 20$)), or cross-infected with the non-native bacterium from *C. bullata* (4 rep ($n = 25$)), *C. gressoria* (5 rep ($n = 13$)), *A. sparsa* (4 rep ($n = 15$)), *A. quinquefasciata* (5 rep ($n = 20$)), and *C. rubiginosa* (5 rep ($n = 20$)). Bars represent the mean proportion infected, whiskers indicate 95% binomial confidence intervals, and dots represent the average of each replicate. Significant differences between treatments are denoted by letters (Fisher's Exact Test, $p = 0.0005$; post hoc comparisons with Holm correction). **B** Symbiont abundance in the foregut symbiotic organs of 5-day-old *C. alternans* larvae, quantified via *16S rRNA* gene copy numbers. Box colors indicate experimental treatments: untreated control (5 rep ($n = 24$)), aposymbiotic (5 rep ($n = 19$)), re-infected with the native bacterium (5 rep ($n = 20$)), or cross-infected with the non-native bacterium from *C. bullata* (4 rep ($n = 25$)), *C. gressoria* (5 rep ($n = 13$)), *A. sparsa* (4 rep ($n = 15$)), *A. quinquefasciata* (5 rep ($n = 20$)), and *C. rubiginosa* (5 rep ($n = 20$)). Lines represent medians, boxes indicate the interquartile range (25–75%), whiskers denote the range, and dots show the average of each replicate. Letters indicate significant differences between treatments (GLM (quasi-Poisson), $df = 7$, $p < 0.001$, $\chi^2 = 119.32$; post hoc comparisons with Bonferroni correction). **C** Larval survivorship to adult eclosion across treatments. Line colors correspond to experimental treatments: dotted lines represent untreated control (7 rep ($n = 61$)) and aposymbiotic (8 rep ($n = 109$)) treatments, while solid lines indicate caplet-free eggs re-infected with the native symbiont (magenta; 7 rep ($n = 32$)) or cross-infected with non-native symbionts from 5 different beetle species (green gradient; *C. bullata*, 7 rep ($n = 64$); *C. gressoria*, 10 rep ($n = 57$); *A. sparsa*, 5 rep ($n = 53$); *A. quinquefasciata*, 7 rep ($n = 31$); and *C. rubiginosa*, 5 rep ($n = 27$)). Beetle survival was monitored until metamorphosis; truncated lines denote surviving individuals censored from the assay. Letters denote significant differences between treatments (Mixed effects Cox regression model, $df = 7$, $p < 0.001$, $\chi^2 = 229.044$; post hoc comparisons with Bonferroni correction). **D, E** Correlations between larval survivorship and pairwise genome-wide comparisons of *S. capleta* average nucleotide identity and collinearity scores. Squares represent the untreated control treatment, while circles indicate caplet-free eggs re-infected with the native symbiont (magenta) or cross-infected with non-native symbionts (green gradient). Spearman correlation coefficients ($r$) and their significance ($p$) are shown in each panel. Source data are provided as a Source Data file.

---

$p < 0.001$; Supplementary Data 2A). In contrast, cross-infected larvae were consistently symbiotic, irrespective of the donor host (Fig. 4A; pairwise comparisons of proportions, *C. bullata* symbiont: $p = 1$, *C. gressoria* symbiont: $p = 1$, *A. sparsa* symbiont: $p = 1$, *A. quinquefasciata* symbiont: $p = 0.24$, *C. rubiginosa* symbiont: $p = 0.67$; Supplementary Data 2A).

Given the consistent cross-infection of *S. capleta* stemming from six donor beetle species, we further quantified symbiont titre to determine whether infection success was reflected in *S. capeta* abundance. Quantitative analysis revealed treatment-dependent variation in symbiont titer (Fig. 4B; GLM (quasi-Poisson), $df = 7$, $p < 0.001$, $\chi^2 = 119.32$; Supplementary Data 2B). *S. capleta* abundances were statistically comparable between larvae harboring their native symbiont and those cross-infected with *C. bullata*, *C. gressoria*, and *A. sparsa* symbionts (Fig. 4B; pairwise comparison, reinfection: $p = 1$, *C. bullata* symbiont: $p = 1$, *C. gressoria* symbiont: $p = 1$, *A. sparsa* symbiont: $p = 0.084$, Supplementary Data 2B). Conversely, larvae cross-infected with *A. quinquefasciata* and *C. rubiginosa* symbionts contained lower symbiont titers than the untreated larvae (Fig. 4B; pairwise comparison, *A. quinquefasciata*: $p < 0.001$, *C. rubiginosa*: $p < 0.001$, Supplementary Data 2B). Quantitative polymerase chain reaction (qPCR) amplification in aposymbiotic individuals was indistinguishable from background (NTC) levels, indicating that the reported copy numbers are likely overestimations and that aposymbiotic larvae are devoid of symbionts, as corroborated by the diagnostic PCR results.

We next quantified how these non-native symbionts shaped the development of *C. alternans* larvae and found that their effects on host survival varied, yielding a gradient of mutualistic outcomes (Fig. 4C; Mixed effects Cox regression model, $df = 7$, $p < 0.001$, $\chi^2 = 229.044$; Supplementary Data 2C). While all cross-infected treatments consistently outpaced aposymbiotic insects (Log-rank tests, $p < 0.001$, Supplementary Data 2C), only two non-native *S. capleta* species completely rescued larval survivorship to levels mirroring to the untreated and re-infected controls (Fig. 4C). These symbionts stem from *C. bullata* (Log-rank tests, $p = 0.7$ and 0.99; Supplementary Data 2C) and *C. gressoria* (Log-rank tests, $p = 0.38$ and 0.76; Supplementary Data 2C), two beetle species that are congeneric (*Chelymorpha*) to the recipient host, *C. alternans* (Fig. 3B). In contrast, the three *S. capleta* species associated with *A. sparsa*, *A. quinquefasciata*, and *C. rubiginosa* could only partially rescue the survivorship of their novel host (Log-rank tests, *A. sparsa*, $p = 0.00029$ and 0.0077; *A. quinquefasciata*, $p = 0.002$ and 0.037; and *C. rubiginosa*, $p = 00013$ and 0.00052, respectively; Supplementary Data 2C), underscoring the diverse developmental consequences of colonization by a non-native symbiont (Fig. 4C). Collectively, this indicates that the observed mortality is only partly

explained by a reduction in symbiont abundance, as seen in the cross-infection with the *A. quinquefasciata* symbiont (Fig. 4B). However, this pattern does not hold for all high-mortality cross-infections, such as those involving the *A. sparsa* symbiont, where symbiont abundance is comparable to those of the native *S. capleta* (Fig. 4B). This variation suggests that host survival following our cross-infections may be influenced by factors beyond symbiont abundance, including reduced symbiont activity and/or the degree of compatibility between host and microbe.

To explore this possibility, we examined whether the mutualistic potential of *S. capleta* in a novel host correlates with its evolutionary relatedness to the native symbiont. *S. capleta* that are closely related to the original symbiont of *C. alternans* were more likely to rescue larval survivorship than more distantly related species (Fig. 4D, E, and Supplementary Data 2D). Using genome-wide pairwise comparisons of average nucleotide identity (ANI) and collinearity as measures of genetic distance, we observed a strong positive correlation between symbiont relatedness and host survivorship (Fig. 4D, E; Spearman's rank correlations: ANI, $r = 0.89$, $p = 0.0071$; collinearity score, $r = 0.92$, $p = 0.0034$; Supplementary Data 2D).

How do different symbiont genotypes influence the survivorship of a novel host? Despite strong metabolic conservation across the *S. capleta* pangenome in categories related to informational processing (e.g., transcription, translation, and replication), different species encode and supplement two iterations of host-beneficial factors to tortoise beetles[36–38]. All genomes of *S. capleta* species sequenced to date encode polygalacturonase (PG), a pectinase that breaks down homogalacturonan, the most abundant pectic substrate[36–38]. However, different symbiont species also produce rhamnogalacturonan lyase (RL), a secondary pectinase that degrades the heteropolymer rhamnogalacturonan I, or α-glucuronidase (AG), a xylanase involved in hemicellulose digestion[36–38].

The six *S. capleta* species used in our study encompass these two configurations of host-beneficial factors: PG and RL, or PG and AG (Supplementary Data 1). The symbiont of *C. alternans*, along with those of *C. gressoria*, *C. bullata*, and *A. sparsa*, all encode PG and AG[36–38]. In contrast, the symbionts of *C. rubiginosa* and *A. quinquefasciata* provide their beetle hosts with PG and RL[36–38]. Since the latter two species only partially restored survivorship in a novel beetle host that typically depends on PG and AG from its native symbiont (Fig. 4C, and Supplementary Data 2C), we hypothesized that differences in host-beneficial factors might explain the observed variation in mutualistic outcomes. However, this is unlikely to be the sole explanation, since the cross-infected symbiont from *A. sparsa* encodes the same set of plant cell wall-degrading enzymes (PG and

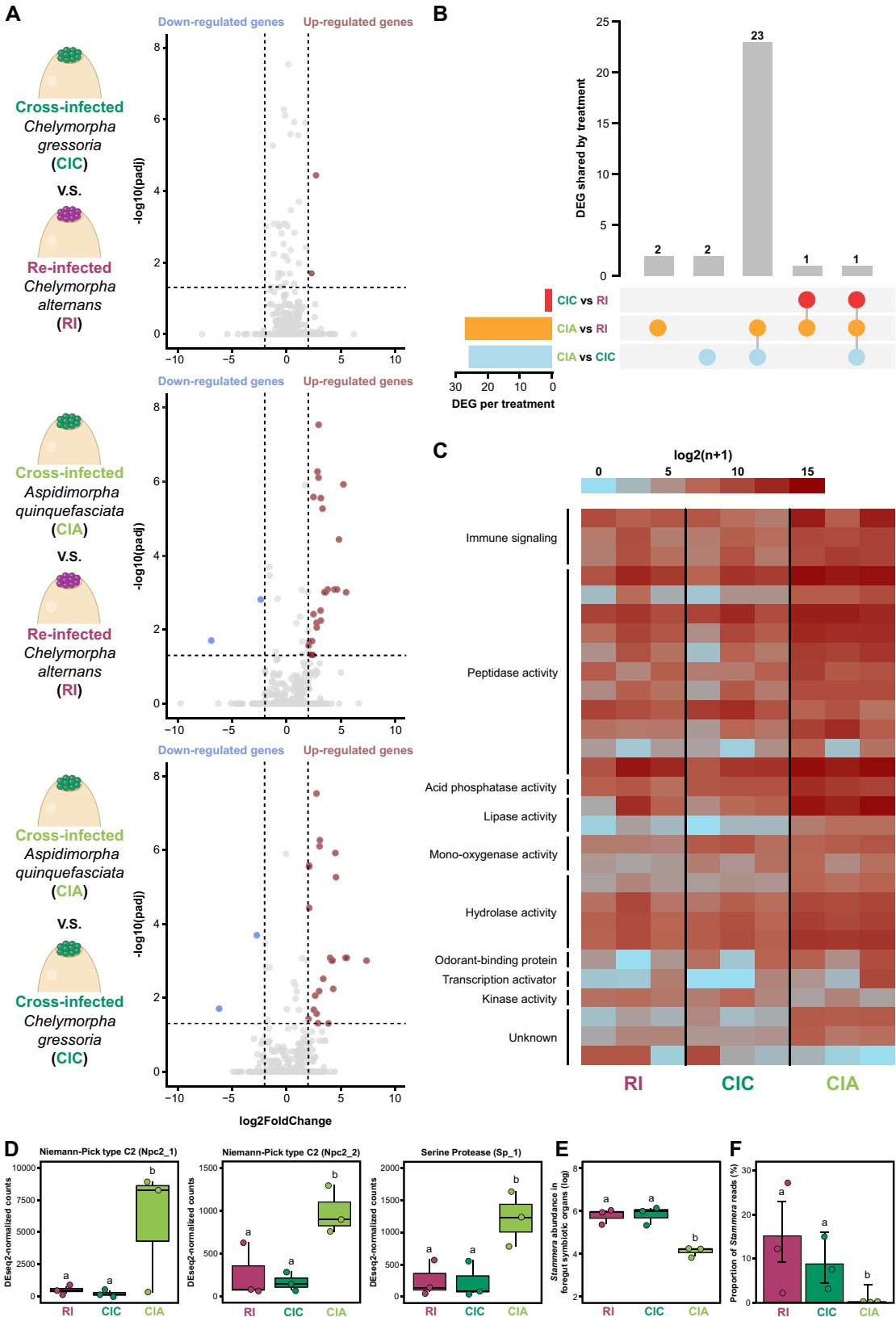

AG) as the native *S. capleta* in *C. alternans* yet fails to restore larval survivorship relative to the untreated and re-infected controls (Fig. 4C, and Supplementary Data 2C). It is conceivable that differences in the expression or regulation of these host-beneficial factors could contribute to the observed variation. Alternatively, the mutualistic potential of *S. capleta* in a novel host may be constrained by other features of its genotype rather than the digestive enzymes it supplements. To explore this further, we investigated the role that partner recognition plays in shaping the symbiotic outcome between tortoise beetles and *S. capleta*. Specifically, we asked: are genetically distant species of *S. capleta* less likely to be recognized as symbionts by their host?

**Fig. 5 | Host response to colonization by a non-native symbiont. A** Differential gene expression in the foregut symbiotic organs of *Chelymorpha alternans* following colonization by native and non-native symbionts. Comparisons include infections by: *Chelymorpha gressoria* symbiont vs. native symbiont (top), *Aspidimorpha quinquefasciata* symbiont vs. native symbiont (middle), and *C. gressoria* symbiont vs. *A. quinquefasciata* symbiont (bottom). Genes with significant expression changes (adjusted $p < 0.05$ and fold change > 2) are highlighted: down-regulated genes in blue and up-regulated genes in red. Three replicates were analyzed per treatment. **B** UpSet plot showing differentially expressed genes (DEGs) across treatments: non-native *C. gressoria* symbiont vs. native symbiont (red), non-native *A. quinquefasciata* symbiont vs. native symbiont (orange), and *C. gressoria* symbiont vs. *A. quinquefasciata* symbiont (blue). Horizontal bars represent the total number of DEGs per treatment, while vertical bars indicate shared and unique DEGs across treatments. **C** Heatmap of log2(x + 1) normalized counts, illustrating host DEG expression patterns in response to colonization by *S. capleta* symbionts. **D** Expression levels of host immune genes after colonization of foregut symbiotic organs by *S. capleta* symbionts. Gene counts were normalized using DESeq2's median of ratios. Bar colors represent experimental treatments: re-infected with the native bacterium (magenta) and cross-infected with the non-native bacterium from *C. gressoria* (dark green) or *A. quinquefasciata* (light green). Medians are shown as lines, boxes represent the 25th to 75th percentiles, whiskers denote the range, and dots indicate the three replicates. Letters show significant differences between treatments (Npc2_1 (NB-GLM, $df = 2$, $p < 0.001$, $\chi^2 = 38.7$), Npc2_2 (NB-GLM, $df = 2$, $p < 0.001$, $\chi^2 = 35.3$), Sp_1 (LM, $p = 0.0012$, $F_{(2,9)} = 56.16$); post hoc comparisons with Bonferroni correction). **E** Symbiont abundance in the foregut symbiotic organs of 5-day-old *C. alternans* larvae, quantified via *16S rRNA* gene copy numbers. Box colors indicate experimental treatments. Lines represent medians, boxes indicate the interquartile range (25–75%), whiskers denote the range, and dots show the three replicates (2 individuals per replicate). Letters indicate significant differences between treatments (LM, $p < 0.001$, $F_{(2,9)} = 42.12$; post hoc comparisons with Bonferroni correction). **F** Proportion of *S. capleta* reads mapped to the symbiont genome relative to total reads mapped (symbiont + beetle genomes). Bars represent the mean proportion of reads, whiskers denote 95% binomial confidence intervals, and dots represent the three replicates. Letters indicate significant differences between treatments (Fisher's Exact Test, $p < 0.001$; post hoc comparisons with Holm correction). Source data are provided as a Source Data file.

## Recognition of an obligate symbiont

Animals that acquire their symbionts horizontally must winnow thousands of encountered microbes down to their selected partner[29,30]. These have shed light on the intricate molecular mechanisms behind symbiont acquisition and partner recognition[22,26,32]. In addition to enriching the environment with beneficial microbes[30,61,62], the host deploys a honed immune system to facilitate symbiont colonization by removing environmental and potentially pathogenic microbes[28,29,31,63–65]. In bean bugs, hundreds of antimicrobial peptides are expressed in the gut, including crypt-specific cysteine-rich peptides[65–67]. These peptides damage the membranes of diverse bacteria but not the crypt-dwelling symbiont, *C. insecticola*, which is resistant[65–67]. Cnidarians, while indiscriminate in their uptake of microalgae from the environment, eject non-symbiotic dinoflagellates through vomocytosis (i.e., engulfment and expulsion by phagocytes)[24]. Similarly, the Hawaiian bobtail squid discriminates for bioluminescent *V. fischeri* through hemocyte binding of its outer membrane[28,63]. Non-symbionts induce an elevated immune response defined by the enrichment of antimicrobial peptides in the symbiotic light organ[29], complementing mechanical and biochemical selective measures[21,34].

Given the varied symbiotic outcomes of *S. capleta* colonization along a genetic gradient (Fig. 4C, D), we investigated whether non-native symbionts trigger a distinct host response. To address this question, we sequenced the transcriptome of the foregut symbiotic organs of *C. alternans* larvae (5-days-old) under three experimental conditions: (a) re-infection with the native symbiont, (b) cross-infection with the symbiont of *C. gressoria* and (c) cross-infection with the symbiont of *A. quinquefasciata* (Fig. 5A). The two non-native symbionts were chosen because of their contrasting effects on host development - while the *C. gressoria* symbiont fully restores larval development in its novel host, the *A. quinquefasciata* symbiont provides only partial rescue (Fig. 4C).

Paired-end, dual RNA sequencing (~30 million reads) was performed on an Illumina NextSeq 2000 and spanning three biological replicates for each treatment (Supplementary Data 3A). Strikingly, only two host genes were differentially expressed between larvae bearing the *C. gressoria* symbiont and insects colonised by their native *S. capleta* (Fig. 5A; Supplementary Data 3B, C, adjusted $p < 0.05$). In contrast, 27 genes were differentially expressed in larvae cross-infected with the *A. quinquefasciata* symbiont compared to the re-infected control (Fig. 5A; Supplementary Data 3B, D, adjusted $p < 0.05$). The *A. quinquefasciata* symbiont, which only partially rescues larval development, triggers a more pronounced host response compared to a non-native symbiont that fully restores it (Fig. 5A).

When the two cross-infected treatments were compared (*A. quinquefasciata* vs. *C. gressoria*), we found 26 genes to be differentially expressed (Fig. 5A; Supplementary Data 3B, E; adjusted $p < 0.05$). Notably, 23 of these genes were also differentially expressed when larvae colonized by their native *S. capleta* were compared to the cross-infected treatment featuring the *A. quinquefasciata* symbiont (Fig. 5B; Supplementary Data 3F, PERMANOVA, $p = 0.025$; Fig. S4 and Supplementary Data 4A). This substantial overlap indicates that the *C. gressoria* symbiont appears to recapitulate the metabolism of its novel host as would the native *S. capleta*.

Genes encoding Niemann-Pick type C2 (NPC2) proteins were highly expressed in cross-infected larvae colonized by the *A. quinquefasciata* symbiont compared to re-infected control and cross-infected larvae harboring the *C. gressoria* symbiont (Fig. 5C, D; Supplementary Data 3B–E; adjusted $p < 0.05$; *Npc2_1*, NB-GLM, $df = 2$, $p < 0.001$, $\chi^2 = 38.7$; *Npc2_2*, NB-GLM, $df = 2$, $p < 0.001$, $\chi^2 = 35.3$; Supplementary Data 4B). NPC2 proteins are known to be involved in sterol homeostasis, steroid biosynthesis, and in innate immunity by modulating the IMD pathway in insects[68–71]. As demonstrated in *Drosophila melanogaster* and the Asian honeybee (*Apis cerana*), NPC2-encoding genes become highly expressed following pathogenic challenges[69,71]. These proteins actively combat microbial infections by directly binding to bacterial and fungal cell walls[69,71], thereby stimulating the expression of antimicrobial peptides[68–70]. It is conceivable that *S. capleta* from *A. quinquefasciata* induces *Npc2* expression because of structural differences in its cell wall compared to the symbionts of *C. gressoria* and *C. alternans*. Supporting this, while the *S. capleta* species from *C. gressoria* and *C. alternans* retained a subset of ligases involved in peptidoglycan biosynthesis (including *murD*, *murE*, D-alanine:D-alanine ligase) and outer membrane assembly proteins (including *BamA* and *PhoE*), the encoding genes are absent from the *A. quinquefasciata* symbiont genome[37,38].

Colonization by the *A. quinquefasciata* symbiont also led to elevated expression of protease-encoding genes, including numerous cysteine cathepsins and a serine protease (Fig. 5C, D and S5, and Supplementary Data 3B–E; adjusted $p < 0.05$; NB-GLM/LM, $p < 0.001$; Supplementary Data 4B). These enzymes are known to have pleiotropic roles in digestion, development, and immune function. In response to bacterial pathogens, serine proteases are shown to activate the Spätzle-mediated Toll-like receptor and the prophenoloxidase pathway in *D. melanogaster* and the mealworm beetle *Tenebrio molitor*[68–70]. Cathepsins, on the other hand, are lysosomal proteases and are classified into over 10 subfamilies based on their primary sequence and enzymatic activity[72]. Many members of the cathepsin superfamily are biologically active and are deployed by insects to

combat pathogenic infections and regulate microbial populations along epithelial cells in the gut[64,73,74].

Our findings suggest that host immunity may contribute to symbiont recognition. Yet they also raise the possibility that a fine-tuned metabolic interplay between host and symbiont, mediated by transcriptional crosstalk, could be essential for proper symbiont growth. The results further reveal upregulation of genes involved in metabolic functions, such as lipase activity (Figs. 5C and S5, and Supplementary Data 3B–E; adjusted $p < 0.05$; Supplementary Data 4B). Such changes could represent a compensatory host response to reduced nutrient access arising from colonization by a metabolically suboptimal symbiont. Consistent with this interpretation, the native (C. alternans) and A. quinquefasciata symbionts differ in their potential metabolic profiles, especially in lipid metabolism[37]. C. alternans and C. gressoria symbionts encode four and three genes, respectively, related to lipid transport and metabolism, whereas the A. quinquefasciata symbiont lacks genes associated with these functions.

We next quantified symbiont abundance using RNA-seq on cDNA and detected significant differences among treatments (Fig. 5E; LM, $p < 0.001$, $F_{(2,9)} = 42.12$; Supplementary Data 4C), consistent with the previously observed proliferation levels of these non-native symbionts (Fig. 4B). S. capleta titers did not differ significantly between larvae colonized by their native symbiont or cross-infected with the C. gressoria symbiont (Fig. 5E; pairwise comparison, $p = 1$; Supplementary Data 4C). In contrast, both groups contained higher symbiont abundance than larvae cross-infected with S. capleta from A. quinquefasciata (Fig. 5E; pairwise comparison, $p < 0.001$ in both cases; Supplementary Data 4C). We also mapped S. capleta reads to measure the relative presence of symbiotic RNA. The proportion of symbiont transcripts differed significantly across the three experimental groups (Fig. 5F; Fisher's Exact Test, $p < 0.001$; Supplementary Data 4D). Larvae colonized by their native symbiont or cross-infected with the C. gressoria symbiont exhibited similar proportions of S. capleta transcripts (Fig. 5F; pairwise comparison of proportions, $p = 0.23$; Supplementary Data 4D), and both groups showed significantly higher proportions than larvae cross-infected with S. capleta from A. quinquefasciata (Fig. 5F; pairwise comparison of proportions, $p = 0.00072$ and $p = 0.023$; Supplementary Data 4D). The pronounced reduction in A. quinquefasciata symbiont abundance suggests that an immune response may have inhibited its proliferation within the foregut symbiotic organ of a novel host. However, it remains unclear whether this decrease reflects bacterial killing or reduced growth in an unfavorable host environment.

Collectively, our findings indicate that (i) non-native S. capleta elicit distinct host responses, reflecting the varied symbiotic outcomes of their colonization in tortoise beetles, where (ii) genetically distant species induce a more pronounced host transcriptional response than those closely related to the native microbe. But given the strict pattern of co-cladogenesis between tortoise beetles and S. capleta[37,38], even among closely related taxa such as C. alternans and C. gressoria, we next asked: is specificity reinforced at a finer resolution later in development?

### Fidelity throughout host development and symbiont transmission

We quantified S. capleta colonization efficiency in adult beetles to determine whether non-native symbionts can persist throughout host development and propagate to future generations. We assessed the symbiotic status of C. alternans hosts that were either re-infected with their native symbiont or cross-infected with S. capleta from C. gressoria at two time points: 5 and 15 days after adult eclosion. In both cases, and despite the extensive transformation and internal reorganization that takes place during metamorphosis, the symbionts persisted throughout pupation and continued to occupy the foregut symbiotic organs of adult beetles (Fig. 6A; GLM (binomial), $df = 1$, $p = 1$, $\chi^2 = 0$; Supplementary Data 4E).

Among adult females, we observed that the ovary-associated glands were successfully colonized by non-native S. capleta, similar to the re-infected control (Figs. 6B–D and S6; GLM (binomial), $df = 1$, $p = 1$, $\chi^2 = 0$; Supplementary Data 4E). The mechanism by which symbionts translocate from the foregut symbiotic organs to the ovary-associated glands is still unknown, but it is conceivable that the non-native symbiont can coopt this pathway to reach the transmission organs of its novel host (Figs. 6B–D and S6). However, despite successfully colonizing the ovary-associated glands, the vertical transmission of S. capleta differed between native and non-native species. While the native symbiont was embedded into the spherical secretions within individual egg caplets, the non-native symbiont was not (Figs. 6E, F and S7).

Eggs laid by cross-infected females contained caplets, but these structures were populated with spheres that were strikingly devoid of S. capleta (Fig. 6F). Diagnostic PCR confirmed the absence of symbionts in the eight egg caplets tested, demonstrating that the non-native S. capleta failed to embed within the maternally secreted spheres. As a result, larvae emerging from these eggs were entirely aposymbiotic (Fig. 6G), in stark contrast to the re-infected treatment (GLM (binomial), $df = 1$, $p < 0.001$, $\chi^2 = 88.98$; Supplementary Data 4E). The ovary-associated glands cross-infected with the C. gressoria symbiont appear to undergo marked morphological remodeling compared to those infected by the native symbiont, from a complex labyrinthine architecture to a simpler conglomerate form between 5 and 15 days (Figs. 6D and S6). How, and whether, such a morphological change affects to the failure to transmit the non-native symbiont is unknown.

Extracellular symbiont transmission is widespread in insects[22]. These routes typically feature a maternal secretions that encapsulate the symbiont[7,22,47,48,75–77], which are then consumed by offspring to initiate infection[22]. For instance, plataspid stinkbugs ensure the vertical transmission of their nutritional symbiont, Ishikawaella capsulata, by embedding it within a proteinaceous capsule[7,48,75]. This capsule is primarily composed of an odorant-binding protein that stabilizes the symbiont's metabolism during extracellular transfer[7], compensating for its highly reduced genome[78]. Similarly, urostylidid stinkbugs propagate their nutritional symbiont, Tachikawaea gelatinosa, through jelly-like secretions that are rich in galactan and essential amino acids[47]. It is conceivable that tortoise beetles rely on similar substrates to produce their S. capleta-bearing spheres (Fig. 1F–H). The embedding properties of these substrates may be specifically tailored to the unique cellular and physiological characteristics of each symbiont, which could explain why non-native S. capleta fail to populate the spherical secretions of their novel host (Fig. 6E, F). Similarly to tortoise beetles, transmission fidelity plays a crucial role for maintaining specificity between beewolves and their defensive ectosymbiont Streptomyces philanthi[79]. While non-native bacteria can colonize beewolf transmission glands, only the native symbiont can successfully embed within the maternal secretion, thereby ensuring the faithful propagation of this partnership[79].

Since many tortoise beetle species are sympatric and often coexist on the same host plant[60], we next examined the infection outcome of multiple S. capleta strains and explored whether local adaptation to the host environment offers a selective advantage to the native symbiont.

### Competition-based selection reinforces specificity

Competition is a strong selective force shaping microbial community structure in a host environment[33,80,81]. The host can promote the dominant growth of locally adapted microbes in open systems like the gut, cuticle, or rhizosphere, by creating a specialized metabolic landscape[33,80,81]. Given recent examples of competition-based selection stabilizing specificity in symbioses[33,82], we investigated whether this process offers an additional mechanism for tortoise beetles to

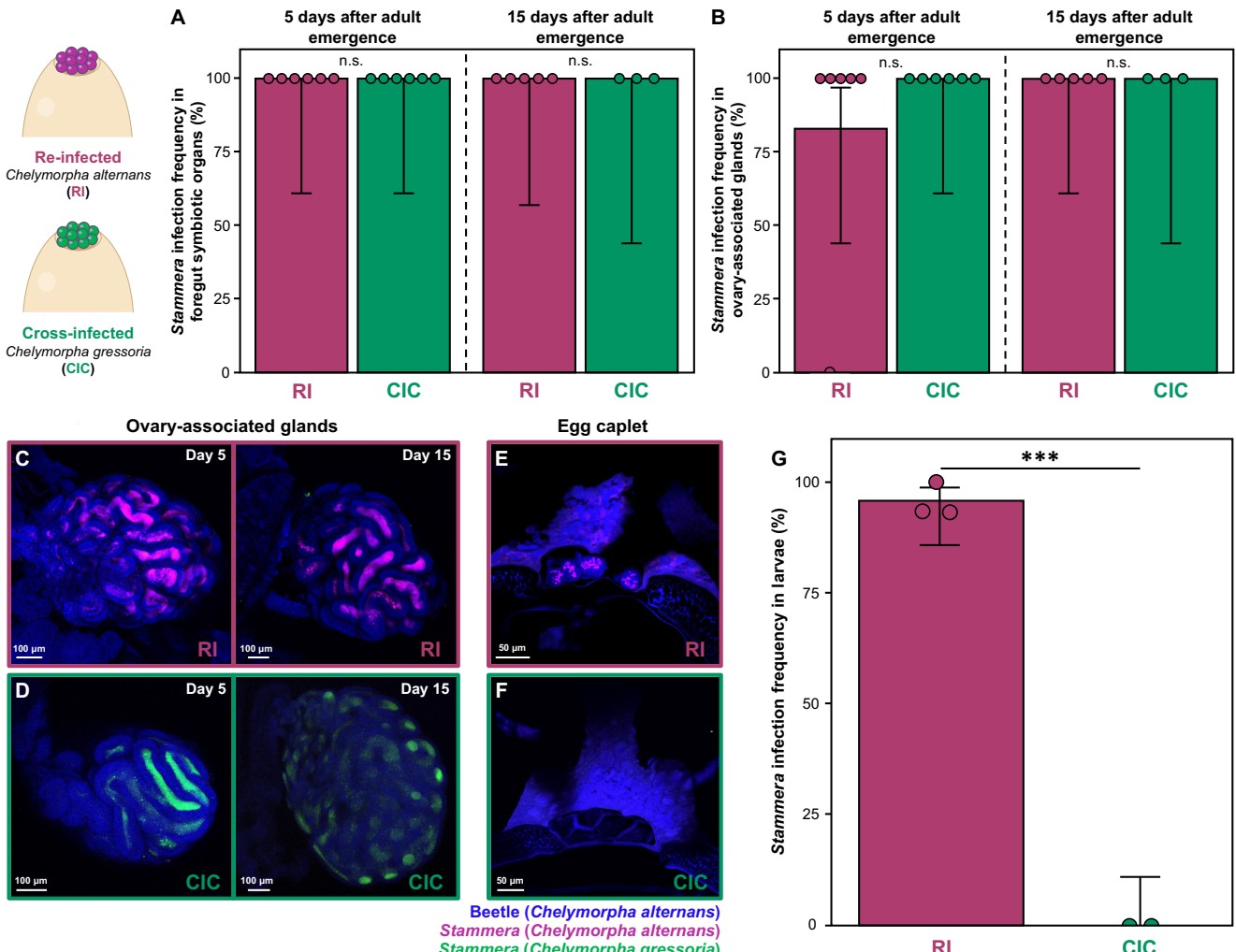

**Fig. 6 | Transmission fidelity ensures propagation of the native symbiont.**
Infection frequencies of native and non-native symbionts in the **A** foregut symbiotic organs and **B** ovary-associated glands of *Chelymorpha alternans* females 5- and 15-days following adult emergence. Bar colors represent experimental treatments: females re-infected with the native symbiont (magenta) or cross-infected with the non-native symbiont (green). Bars represent the mean proportion infected, whiskers indicate 95% binomial confidence intervals, and dots represent replicates (re-infected at 5 days, $n = 6$; re-infected at 15 days, $n = 5$; cross-infected at 5 days, $n = 6$; cross-infected at 15 days, $n = 3$). Non-significant differences in the interaction between treatment and development time are denoted as "n.s." (foregut symbiotic organs: GLM (binomial), $df = 1$, $p = 1$, $\chi^2 = 0$; ovary-associated glands: GLM (binomial), $df = 1$, $p = 1$, $\chi^2 = 0$). **C, D** Fluorescence in situ hybridization (FISH) of whole-mounts of ovary-associated glands from *C. alternans* females re-infected with the native symbiont (magenta) or cross-infected with the non-native symbiont (green), visualized at 5- and 15-days following adult emergence. Each image is from a single individual and is shown for illustrative purposes (see Fig. S6). Scale bars represent 100 μm. **E, F** FISH of longitudinal egg sections laid by *C. alternans* females either re-infected with the native symbiont (magenta) or cross-infected with the non-native symbiont (green). Each image is representative of one individual; similar results were observed in two replicates per treatment ($n = 1$–4 eggs per replicate; see Fig. S7). Dark blue autofluorescence is observed in egg caplets. Scale bars represent 50 μm. Probes used: *S. capleta* from *C. alternans* (magenta: 16S rRNA), *S. capleta* from *C. gressoria* (green: 16S rRNA), and *C. alternans* host (dark blue: 18S rRNA). All symbiont-targeting probes are specific and used simultaneously during hybridization. **G** Infection frequency of native and non-native symbionts in larvae emerging from eggs laid by re-infected ($n = 48$) or cross-infected ($n = 30$) females. Bars represent the mean proportion infected, whiskers indicate 95% binomial confidence intervals, and dots represent the average of each replicate. Significant differences between symbionts are marked with asterisks (GLM (binomial), $df = 1$, $p < 0.001$, $\chi^2 = 88.98$). Source data are provided as a Source Data file.

promote the competitive advantage of their native *S. capleta* during mixed infections.

Using *C. alternans* as a host, we conducted dual-infection assays by introducing two *S. capleta*-bearing spheres to caplet-free eggs: one containing the native symbiont, and the other collected from the congeneric beetle, *C. gressoria* (Fig. 7A). Both symbionts successfully colonized the foregut symbiotic organs of 5-day-old larvae, demonstrating that tortoise beetles can support simultaneous infections by multiple *S. capleta* strains (Figs. 7Bi–iv and S8). We validated these findings by quantifying symbiont abundance throughout host

development (Fig. 7C; LM, $p < 0.001$, $F_{(4,24)} = 83.61$; Supplementary Data 4F).

In newly hatched larvae, native and non-native *S. capleta* initially co-colonized the host at similar titers (Fig. 7C; post-eclosion: day 1 (pairwise comparisons of contrasts, $p = 0.99$), day 5 (pairwise comparisons of contrasts, $p = 0.41$); Supplementary Data 4F). However, symbiont abundance diverged later in larval development (Fig. 7C; 12 days post hatching, pairwise comparisons of contrasts, $p = 0.0024$; Supplementary Data 4F), despite encoding a near-identical (-97%) gene set[38]. Both symbionts share 226 genes but differ in only eight.

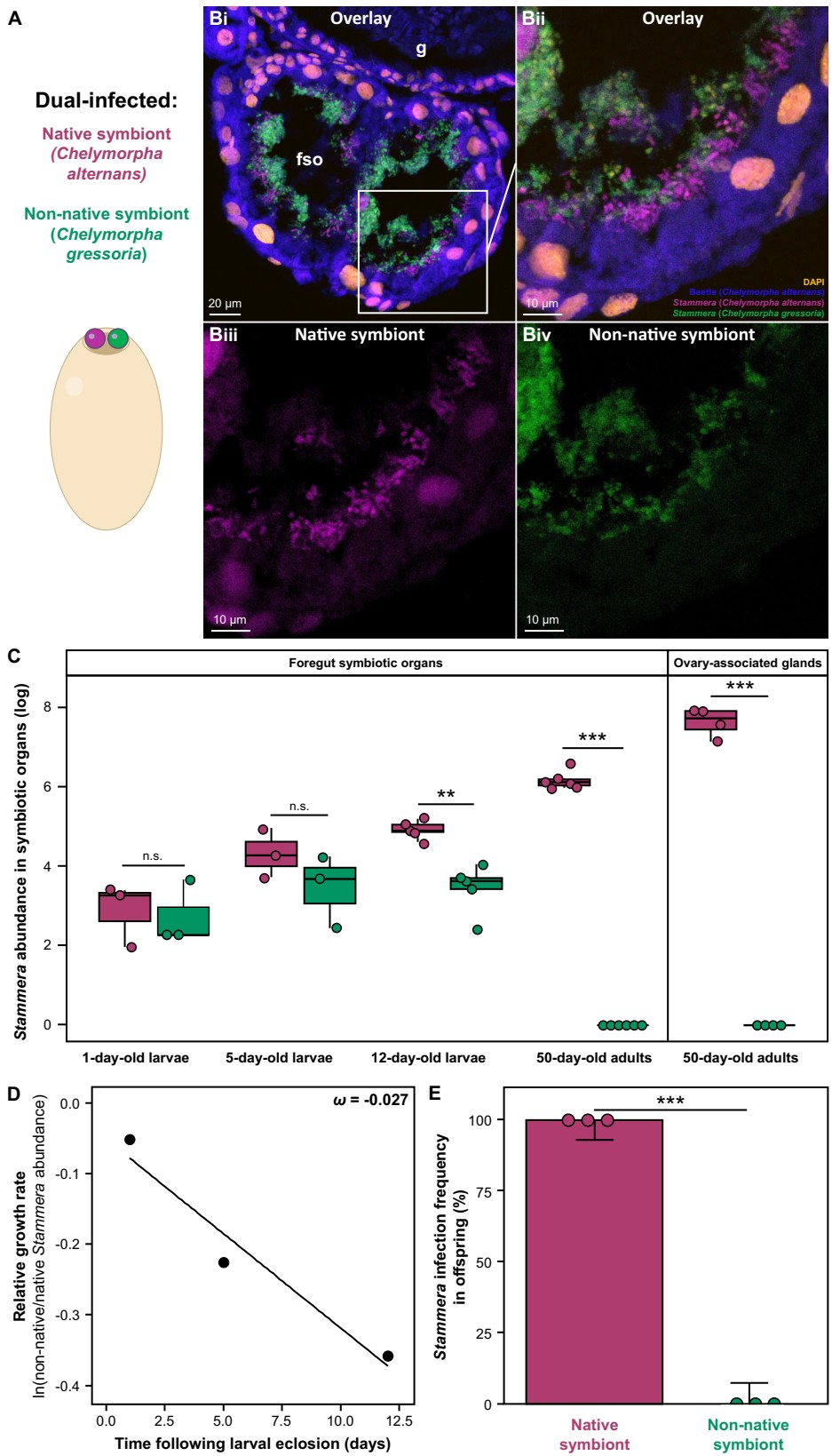

The *C. alternans* symbiont possesses genes associated with translation and energy metabolism, whereas the *C. gressoria* symbiont encodes genes involved in amino acid and cell wall biosynthesis, reflecting differences in metabolic specialization (Table S1). The variations in symbiont population growth may be attributed to these specific genetic differences and/or to variations in the regulation of

gene expression that led to distinct physiological capacities. To estimate the growth rate of the non-native *S. capleta* relative to the native one during larval development, we calculated a selection coefficient. This yielded a coefficient of −0.027 (Fig. 7D), indicating a selective disadvantage for the non-native microbe. A second, independent method leveraging Malthusian parameters corroborated

**Fig. 7 | Competition reinforces specificity between tortoise beetles and *Stammera capleta*. A** Illustration of dual-infected *Chelymorpha alternans* embryos. The native symbiont corresponds to *S. capleta* from *C. alternans*, while the non-native symbiont stems from *Chelymorpha gressoria*. **B** Fluorescence in situ hybridization (FISH) cross-section of foregut symbiotic organs in 5-day-old dual-infected *C. alternans* larvae. **B**ii–iv correspond to a close-up view of the foregut symbiotic organ. **B**i, ii correspond to the merged channels images, while **B**iii corresponds to the native symbiont channel image, and **B**iv to the non-native channel image. The image is from a single individual and is shown for illustrative purposes (see Fig. S8). Probes: *S. capleta* from *C. alternans* (magenta: 16S rRNA), *S. capleta* from *C. gressoria* (green: 16S rRNA), *C. alternans* host (dark blue: 18S rRNA), and DAPI-stained DNA (orange). Symbiont-targeting probes are highly specific and used simultaneously during hybridization. Scale bars are included for reference. **C** Population dynamics of native and non-native symbionts in the symbiotic organs of dual-infected *C. alternans* beetles during development, quantified via *groEL* gene copy numbers. Box colors indicate either the native symbiont (magenta) or non-native symbiont (green). Lines represent medians, boxes indicate the interquartile range

(25–75%), whiskers denote the range, and dots show replicates (1-day-old larvae, $n = 3$; 5-day-old larvae, $n = 3$; 12-day-old larvae, $n = 5$; 50-day-old adults in the foregut symbiotic organs, $n = 6$; 50-day-old adults in the ovary-associated glands, $n = 4$). Stars or "n.s." indicate significant or non-significant differences between symbionts (LM, $p < 0.001$, $F_{(4,24)} = 83.61$; post hoc comparisons using least-squares means with Tukey correction). **D** Relative growth rate of non-native symbionts compared to native symbionts in the foregut symbiotic organs of dual-infected *C. alternans* larvae. The slope is used to calculate the selection coefficient ($\omega$), shown overlaid on the plot. **E** Infection frequencies of native and non-native *S. capleta* in larvae derived from dual-infected *C. alternans* adults. Box colors indicate the experimental treatment: larvae from eggs laid by dual-infected *C. alternans* beetles. Three replicates were collected per treatment ($n = 48$). Bars represent the mean proportion infected, whiskers represent 95% binomial confidence intervals, and dots represent the average of each replicate. Significant differences between symbionts are indicated with stars (GLM (binomial), $df = 1$, $p < 0.001$, $\chi^2 = 133.08$). g gut, fso foregut symbiotic organs. Source data are provided as a Source Data file.

this result, resulting in a similarly negative selection rate of −0.025 for the non-native species.

By adulthood, quantification of symbiont titers revealed the complete loss of the non-native species, while the native *S. capleta* continued to stably colonize both the foregut symbiotic organs and the ovary-associated glands, underscoring the role of microbe-microbe competition in reinforcing specificity between tortoise beetles and *S. capleta* (Fig. 7C; pairwise comparisons of contrasts: foregut symbiotic organs, $p < 0.001$; ovary-associated glands, $p < 0.001$; Supplementary Data 4F). This contrasts the infection dynamics observed when the *C. gressoria* symbiont colonized *C. alternans* alone—where it successfully persisted in both the adult foregut symbiotic organs and ovary-associated glands of its novel host (Fig. 6A–D). Reflecting this, offspring of dual-infected beetles were exclusively colonized by the native symbiont (Fig. 7E; GLM (binomial), $df = 1$, $p < 0.001$, $\chi^2 = 133.08$; Supplementary Data 4G).

Our findings show that once both symbionts colonize the foregut symbiotic organs, the native population rapidly supplants the non-native one, despite their near-functional equivalence and effects on host fitness (Fig. 4) and metabolic response (Fig. 5). This indicates that the native bacterium is better adapted to its host and reinforces the hypothesis that co-evolutionary history is a key driver of fidelity. We further observed spatial heterogeneity in symbiont distribution across the symbiotic organs. The native bacterium predominantly occupies regions adjacent to the epithelium, whereas the non-native bacterium remains restricted to the lumen (Fig. 7Bi-iv and S8). The observed distribution may partly explain the native symbiont's competitive advantage, potentially through greater local adaptation and enhanced access to host-derived nutrients. We note that a reciprocal dual-infection experiment in *C. gressoria* would help rule out the alternative possibility that the *C. gressoria* symbiont is intrinsically a weaker competitor due to differences in its metabolic relative to *S. capleta* associated with *C. alternans*.

Overall, our findings align with recent studies on bean bugs[33] and mice[82], demonstrating that native symbionts consistently outcompete non-native microbes within host environments. This highlights the selective advantage of local adaptation, which can further strengthen specificity within symbiotic partnerships.

Collectively, we conducted experimental cross-infections in aposymbiotic eggs of the tortoise beetle *C. alternans* using *S. capleta* derived from five additional Cassidinae species. Our results demonstrate that larval survival is restored following cross-infections, but only when the non-native symbionts are closely genetically related to the native one. Consistent with this, the positive correlation between symbiont genetic distance and host survivorship highlights the strong influence of symbiont genotype on host development.

Our findings also demonstrate that while obligate, heritable endosymbionts can successfully establish and fully rescue development in a novel host, long-term persistence and successful vertical transmission are maintained by a robust framework of partner specificity. The introduction of a genetically distant, non-native symbiont triggered a strong transcriptional response in the host, including activation of immune genes. This was accompanied by a decrease in symbiont population, suggesting host discrimination between native and non-native partners. When both were introduced simultaneously, the native symbiont progressively outcompeted the non-native one and was exclusively transmitted to the next generation.

Together, these results highlight how the multilayered interplay of host recognition, transmission fidelity, and competitive exclusion acts to ensure that only the native symbiont persists, securing propagation to the next host generation. This selective advantage highlights the evolutionary fine-tuning of the *S. capleta*-tortoise beetle partnership, where co-diversification and local adaptation appear to reinforce a highly specific, heritable symbiosis. Testing whether these mechanisms act in concert to preserve specificity in other vertically transmitted, co-diversified systems would be valuable given the continued development of experimental approaches to clear and reconstitute endosymbionts, as demonstrated in aphids[83], weevils[18], and plataspid stinkbugs[84]. Moreover, the application of microinjection into the host hemolymph to reproduce the mounted immune response is a worthy area for future study.

Our results align with and extend findings from horizontally acquired symbioses, such as the *V. fischeri*-Hawaiian bobtail squid and *C. insecticola*-bean bug systems, where host-driven selection - via immune filtering, niche restriction, and competitive exclusion - ensures the establishment of preferred microbial partners. In the squid, hemocyte-mediated recognition and antimicrobial peptide expression help winnow *Vibrio* from a diverse seawater microbiome[21,26,28,29,34], while the bean bug's gut crypts physically and chemically promote the retention of *Caballeronia* over environmental bacteria[19,20,30,31,33]. The *S. capleta*-tortoise beetle model demonstrates that even vertically transmitted endosymbionts are subject to similar competitive and immunological pressures, blurring the distinction between horizontal and vertical symbiont selection mechanisms. Future research comparing obligate, facultative, and environmentally acquired symbioses could uncover shared genetic and molecular pathways that mediate partner fidelity, with implications for the establishment and maintenance of complex microbial communities across ecological and evolutionary scales[10,82].

## Methods

This study complies with Nagoya Protocol regulations in the Republic of Panama and France per the following permits: ARG-027-2023 and TREL2302365S/693 for the collection and use of biological materials.

### Insect rearing

Tortoise beetles were maintained in a greenhouse at the Max Planck Institute for Biology in Tübingen, Germany. The insects were reared in mesh containers along with their host plants: *Ipomoea batatas* for *C. alternans*, *C. bullata*, *C. gressoria*, and *A. quinquefasciata* beetles; *Merremia umbellata* for *A. sparsa* beetle; and *Cirsium oleraceum* for *C. rubiginosa* beetle (Supplementary Data 1). Experimental treatments were maintained in climate chambers at a constant temperature of 26 °C, humidity of 60%, and long light regimes (14.30 h/9.30 h light/ dark cycles).

### Whole-mount fluorescence in situ hybridization

To visualize *S. capleta* in *C. alternans* eggs and adults, whole-mount fluorescence in situ hybridization (FISH) was performed, as previously described[85,86]. Eggs and a single dissection from an ovipositing female, containing the foregut, midgut, hindgut, and ovaries, were fixed in Carnoy's solution (ethanol: chloroform: acetic acid = 6: 3: 1) at room temperature overnight. After thorough washing with 100% ethanol, the samples were bleached in an alcoholic 6% $H_2O_2$ solution to reduce autofluorescence and incubated at room temperature until they were decolorized. Samples were thoroughly washed with 100% and then 70% ethanol and rehydrated with PBSTx containing 0.3% Triton X-100. They were subsequently incubated with hybridization buffer (20 mM Tris-HCl [pH 8.0], 0.9 M NaCl, 0.01% sodium dodecyl sulfate, 30% formamide) containing 100 nM of each of the probes overnight. The female internal organs were hybridized with the oligonucleotide probes Atto550-SAL227 targeting the 16S rRNA sequence of *S. capleta* from *C. alternans* and Atto488-EUKCA164 targeting the 18S rRNA sequence of *C. alternans* (Supplementary Data 5A). Eggs were hybridized with the oligonucleotide probes Atto550-SAL227 targeting the 16S rRNA sequence of *S. capleta* from *C. alternans* and Atto488-EUK1195 targeting the 18S rRNA sequence of all eukaryotes (Supplementary Data 5A). All probes were dually labeled with fluorescent dyes at 5′ and 3′ ends. Samples were washed thoroughly with PBSTx, mounted using ProLong® Gold antifade mounting media (Thermo Fisher Scientific, USA), and observed using an LSM 980 NLO confocal microscope (Zeiss, Germany).

### Fluorescence in situ hybridization on paraffin sections

To localize *S. capleta* in foregut symbiotic organs of the six tortoise beetle species and in *C. alternans* eggs, FISH was applied on paraffin sections. Adult female foregut symbiotic organs and eggs were dissected and fixed in Carnoy's solution at room temperature overnight. Dehydration was performed by incubating the samples in 70 and 80% ethanol for 10 min each, followed by three 10-min incubations in 96 and 100% ethanol. Samples were gradually transferred into paraffin by passing through three incubations of Roti®-Histol (CarlRoth, Germany) at room temperature (two steps of 40 min and one step overnight), followed by incubations at 60 °C in Roti®-Histol: paraffin (1:1 v/v) for 60 min and paraffin (Paraplast plus, Leica, Germany) (three steps of 60 min and one step overnight). The paraffin-embedded samples were cross-sectioned at 10 μm using a microtome and mounted on poly-L-lysine-coated glass slides (Epredia, Germany). Paraffin sections were dried at room temperature overnight and incubated at 60 °C for 1 h to improve tissue adherence. They were dewaxed with Roti®-Histol in three consecutive steps for 10 min each, followed by ethanol 100% for 10 min. Slides were then dried at 37 °C for 30 min. Foregut symbiotic organs were hybridized with the oligonucleotide probe Cya5-SCA600 targeting the 16S rRNA sequence of *S. capleta* from 52 cassidines (Supplementary Data 5A). Eggs were hybridized with the

oligonucleotide probes Atto550-SAL227 targeting the 16S rRNA sequence of *S. capleta* from *C. alternans* and Atto488-EUKCA164 targeting the 18S rRNA sequence of *C. alternans* (Supplementary Data 5A). Probes, dually labeled with the fluorescent dye at their 5′ and 3′ ends, were dissolved at 900 nM in the hybridization buffer containing 35% formamide (v/v), 900 mM NaCl, 20 mM Tris-HCl pH 7.8, 1% blocking reagent for nucleic acids (v/v) (Roche, Switzerland), 0.02 SDS (v/v), and 10% dextran sulfate (w/v). Hybridization was performed at 46 °C for 4 h. Sections were then transferred to fresh 48 °C washing buffer for 15 min [70 mM NaCl, 20 mM Tris-HCl pH 7.8, 5 mM ethylenedia-minetetraacetic acid pH 8.0, and 0.01% SDS (v/v)] followed by room-temperature washes in PBS (20 min) and Milli-Q® water (1 min). They were counterstained with DAPI (5 μg/ml) for 10 min at room temperature, dipped in Milli-Q® water, ethanol 100%, and dried at 30 °C for 10 min. Slides were mounted using the ProLong® Gold antifade mounting media (Thermo Fisher Scientific, USA), cured overnight at room temperature, and visualized using an LSM 980 NLO confocal microscope (Zeiss, Germany).

### Experimental symbiont exchange across beetle species

Egg masses containing 30 or more eggs with well-defined caplets were collected from *C. alternans* mating pairs to avoid pseudo-replication. Egg masses were then separated into different experimental treatments depending on the experiment: untreated control, aposymbiotic, re-infected with the native symbiont, and cross-infected with symbionts from a different beetle species. Aposymbiotic individuals were generated by removing caplets from eggs using sterile dissection scissors and without piercing the developing embryo[36,39,40]. Re-infection with the native bacterium was performed by eliminating caplets from eggs and reapplying all enclosed spheres to the anterior pole of the egg, as previously described[39,40]. Cross-infections with non-native symbionts were achieved by removing caplets from eggs and reapplying all enclosed spheres collected from eggs of different beetle species to the anterior pole of *C. alternans* eggs. As with the re-infection procedure, spheres were obtained from eggs of the same mass for each replicate.

To test whether *S. capleta* can be reciprocally exchanged across tortoise beetles, whole-mount FISH was performed on entire embryos and larval foregut symbiotic organs of two beetle species *C. alternans* and *C. gressoria*, as described above. For each beetle species, two to four egg masses were collected from separate females and divided into four experimental treatments: untreated control, eggs with caplets removed (aposymbiotic), eggs with caplets removed but re-supplied with their original symbiont-bearing spheres, and eggs with caplets removed but supplied with spheres collected from different host species. A near-emergence embryo (-10.5 days following oviposition) and 5-day-old larvae were collected after experimental manipulation for each replicate. Samples were placed in 70% ethanol to remove the embryo legs and scoli and dissect the larval foregut symbiotic organs using sterilized scissors. The following oligonucleotide probes were used for in situ hybridization: Atto550-SAL227 targeting the 16S rRNA sequence of *S. capleta* from *C. alternans*, Cy5-SCG224 targeting the 16S rRNA sequence of *S. capleta* from *C. gressoria*, Atto488-EUKCA164 targeting the 18S rRNA sequence of *C. alternans*, and Atto488-EUKCG167 targeting the 18S rRNA sequence of *C. gressoria* (Supplementary Data 5A). Symbiont-targeting probes are specific and used simultaneously during hybridization. The formamide curve generator tool in mathFISH was used to analyze the formamide curves and check the specificity of each probe. All probes were dually labeled with fluorescent dyes at 5′ and 3′ ends. Controls consisted of the two beetle species infected with their native symbionts (Fig. S1).

### Phylogenetic inference

Host mitochondrial genomes were extracted from metagenomic assemblies and aligned using MUSCLE (v3.8.1551)[87], and as described

by Garcia-Lozano and colleagues[38] (Supplementary Data 5B). A concatenated alignment of 15 mitochondrial genes (13 protein-coding genes + two ribosomal rRNA genes) was partitioned to assign the most appropriate substitution model to each gene using PartitionFinder2 (branchlengths=unlinked, models = GTR+G and GTR+I+G, model_selection = bic; v2.1.1)[88]. Phylogenetic analyzes were performed using Maximum Likelihood (ML) in RAxML-NG (v1.2.0)[89] (ngen=1000). Members of Spilopyrinae (*Spilopyra sumptuosa*) and Eumolpinae (*Bromius obscurus*, *Colaspidea* sp., and *Paria* sp.) subfamilies from the Chrysomelidae were used as outgroups for this analysis.

To infer the relationships between the 6 corresponding *S. capleta* symbionts, 61 single-copy core genes identified by anvi'o (v8.1-dev)[90] and present in all symbionts were extracted from each *S. capleta* genome and aligned using MUSCLE (v3.8.1551)[38,87]. Concatenated alignments of these genes and four outgroups (*Escherichia coli*, *B. aphidicola*, *Candidatus* Ishikawaella capsulata, and *V. fischeri*) were included to construct a ML phylogeny using RAxML-NG (v1.2.0)[89] (ngen=1000) (Supplementary Data 5B). The best-fit substitution model was selected using PartitionFinder2 (branchlengths=unlinked, models = GTR+G, GTR+I+ G, model_selection = bic; v2.1.1)[88]. The tree reconciliation software eMPRess GUI (v1.0.)[91] was applied to investigate the evolutionary relationship between the 6 beetle species and their symbionts. This software reconciles symbiont and host trees using the DuplicationTransfer-Loss model. ML trees were used as input and the analysis was conducted using the following parameters: duplication cost = 1, transfer cost = 1, and loss cost = 1. The significance of reconciliation between host and symbiont tree was calculated by randomizing the tips of the branches and then, re-calculating the cost to reconcile the phylogenies. Congruent phylogenies are obtained when the original cost of reconciliation is less than expected by chance ($p < 0.01$).

### *S. capleta* genetic distance

A dual approach was used to evaluate sequence similarity and structural gene organization across symbiont genomes[38]. The ANI between *S. capleta* genomes was calculated using PyANI (v2.4.0)[92], and as implemented in anvi'o (v8.1-dev)[90] to assess symbiont relatedness. Gene collinearity across different *S. capleta* lineages was analyzed with MCScanX (v1.0)[93] and SynTracker (v1.1.3)[94]. Collinearity scores were then calculated using the collinearity scripts from Nowell et al.[95]. To adapt SynTracker for the comparison of different species we used modified parameters (minimal blast identity = 70%, minimal coverage=70%). SynTracker comparison was carried out with the collection of six *S. capleta* genomes as potential references. The Average Pairwise Synteny score was calculated using all comparable regions, for each genome pair. Next, the score for each genome-pair was multiplied by the number of comparable regions, normalized by the length of the smaller genome in each genome-pair, and transformed to fit a scale of 1–100.

### Experimental manipulation to quantify the impact of non-native *S. capleta* on beetle development

To estimate whether non-native symbionts can successfully colonize the foregut symbiotic organs of *C. alternans* and restore its survival, egg masses were collected from different female beetles and divided into eight experimental treatments: untreated control, aposymbiotic eggs, eggs whose caplets were removed but re-infected with the native bacterium, or eggs whose caplets were removed but cross-infected with *S. capleta*-bearing spheres collected from 5 different beetle species (*C. bullata*, *C. gressoria*, *A. sparsa*, *A. quinquefasciata*, and *C. rubiginosa*).

Symbiont infection frequencies were validated across all treatments using diagnostic PCR. Five days after hatching, foregut symbiotic organs of larvae were dissected, and DNA was extracted using the E.Z.N.A.® Insect DNA Kit from Omega Bio-tek. PCR primers targeting either specific *S. capleta* symbionts (*groEL* gene) or all symbionts collectively (*16S rRNA* gene) were used to verify the symbiotic status of the host beetles (Supplementary Data 5C). For cross-infected individuals, a *groEL* primer pair specific to the native symbiont was used to confirm its successful removal. The primer pairs were designed using Primer3 and Geneious software and synthesized by Sigma-Aldrich under standard conditions. PCR products were sequenced to confirm primer specificity in vitro. Additionally, a specific primer pair for the *CO1* gene of *C. alternans* was used as a control for the DNA extraction (Supplementary Data 5C). Diagnostic PCR was conducted on an Analytik Jena Biometra TAdvanced Thermal Cycler using a final volume of 20 μl containing 1 μl of DNA template, 0.5 μM of each primer, and 2× DreamTaq PCR Master Mix. The following cycle parameters were used: 5 min at 95 °C, followed by 34 cycles of 95 °C for 30 s, 54.7 to 62.5 °C (depending on the primer pair, Supplementary Data 5C) for 30 s, 72 °C for 1 min and a final extension time of 2 min at 72 °C. Treatments were carried out in different experimental blocks. Four to five replicates were performed depending on treatment (i.e., $n = 13$-to-25 larvae).

The impact of the cross-infection procedure on symbiont abundances was assessed by qPCR using an Analytik Jena® qTOWER³ G cycler. DNA samples used previously served as templates. Symbiont abundance was estimated by quantifying the copy number of the *S. capleta 16S rRNA* gene, targeting all Cassidinae species studied, using the following primer pair: TTGACCGCCTGGGGAGTA and TTCGCGTTGCATCGAATTAAAC (Supplementary Data 5D). The primer pair was designed using Geneious software and synthesized by Sigma-Aldrich under standard conditions. Verification of primer specificity was conducted in silico by comparison with reference sequences of all symbionts. qPCR reactions were performed in a final volume of 25 μl, containing 1 μl of DNA template, 2.5 μl of each primer (10 μM), 6.5 μl of autoclaved distilled $H_2O$, and 12.5 μl of the Platinum SYBR Green qPCR SuperMix-UDG (Thermo Fisher Scientific). Cycling conditions consisted of 3 min activation at 95 °C, followed by 45 cycles of 95 °C for 15 s, 62 °C for 45 s, 72 °C for 20 s. A melting curve analysis was conducted from 60 to 95 °C, increasing the temperature by 1° every 30 s. Gene copy numbers were estimated from standard curves (tenfold dilution series from $10^{-1}$ to $10^{-8}$ ng/μl) generated from purified PCR products, whose DNA concentrations were measured with a NanoDrop™ 1000 spectrophotometer. Since a slight amplification signal was detected in the negative controls (NTCs), the gene copy numbers were corrected by subtracting the average copy number obtained for the NTCs. Treatments were carried out in separate experimental blocks, with four to five replicates per treatment ($n = 13$-to-25 larvae).

Emerged larvae were observed daily, and survival until adulthood was recorded to assess the impact of non-native symbionts on larval survivorship. Treatments were carried out in different experimental blocks. Five-to-ten replicates were performed depending on treatment (i.e., $n = 27$ to 109 larvae). The relationship between the percentage of survivors after adult emergence and symbiont genetic distance (ANI or collinearity score) was also tested.

### RNA sequencing

The transcriptome of foregut symbiotic organs of 5-day-old *C. alternans* larvae was sequenced across three experimental treatments: (a) re-infection with the native symbiont, (b) cross-infection with the symbiont of *C. gressoria* and (c) cross-infection with the symbiont of *A. quinquefasciata*. Three egg masses were collected from separate females and divided into these experimental treatments. Two larvae from each replicate were sampled 5 days after hatching, and foregut symbiotic organs were dissected using sterile scissors and snap-frozen in liquid nitrogen. RNA extraction was performed for each sample immediately after collection using the QIAGEN RNeasy Mini Kit according to the protocol 4: Enzymatic Lysis and Proteinase K

Digestion of Bacteria starting from step 7. This protocol is included in the RNAprotect® Bacteria Reagent Handbook from Qiagen and as referenced in Ref. [38]. Total RNA was quantified using the Qubit™ RNA High Sensitivity (HS) kit. The two samples were pooled to obtain 40 ng of total RNA, which was used as input to prepare nine RNA sequencing libraries (3 biological replicates). Ribosomal RNA was depleted from total RNA using the NEBNext® rRNA Depletion Kit (Human/Mouse/Rat). Libraries were constructed using the NEBNext® Ultra™ Directional II RNA Library Prep kit, and their size was confirmed in a 2100 Bioanalyzer system using the Agilent Technologies High Sensitivity DNA Kit. Sequencing of the final libraries was performed on an Illumina NextSeq 2000 system at the Max Planck Institute for Biology (Tübingen, Germany) using paired-end chemistry ($2 \times 100$bp) with a depth of ~30 million reads.

Adapter removal and quality filtering of raw reads were performed in Trimmomatic (v0.36)[96] and FastQC (v0.12.0)[97]. Filtered reads were mapped to the *C. alternans* genome (Garcia-Lozano et al. personal communication) using HISAT2 (v2.2.0)[98]. Gene counts were obtained using htseq-count (v0.11.5)[99] (stranded=reverse -r pos). After applying a filter, read counts were normalized by the DESeq2's median of ratios established in the *DESeq2* package[100]. The likelihood ratio test (LRT), implemented in *DESeq2*, was used to test for differences in host gene expression across treatments. Host genes were considered significantly differentially expressed at adjusted *p*-value (*p* adj) <0.05 and a fold-change > 2. Volcano plots were generated using the ggplot2[101] and dplyr[102] packages for visualizing transcriptome data. We compared the differentially expressed gene (DEG) profile between treatments by testing for significant clusters using a permuted multivariate analysis of variance (PERMANOVA) and applying the function vegan::adonis(). The batch effect was removed using the *LIMMA* package[103]. DEGs were further annotated for immune functions using the *C. alternans* immune gene reference set (Garcia-Lozano et al. personal communication), which was derived from the 4IN database (Innate Immunity Genes in Insects: http://bf2i300.insa-lyon.fr:443/home). A heatmap of the $\log_2(x + 1)$ normalized counts was then constructed to illustrate host DEG expression patterns in response to colonization by *S. capleta* symbionts. Analyzes were performed in R v. 3.5.3[104].

Symbiont abundances across treatments were quantified by qPCR on an Analytik Jena® qTOWER³ G cycler. cDNA generated from the corresponding RNA samples were used as templates. For cDNA synthesis, 10 ng of total RNA per sample was reverse-transcribed using the RevertAid First Strand cDNA Synthesis Kit (Thermo Fisher Scientific) with oligo(dT) primers. qPCR was performed with the same primer pair (*16S rRNA*) and conditions as described earlier, except that the standard curve was constructed from a tenfold serial dilution of cDNA ($10^{-1}$ to $10^{-8}$ ng/µl). Three replicates were performed per treatment ($n = 6$ larvae). In addition, the proportion of *S. capleta* symbionts within each treatment was estimated as the ratio of RNA-seq reads mapped to the genome of the target symbiont relative the total number of mapped reads (i.e., reads aligned to both the symbiont and host genomes).

### Persistence of non-native *S. capleta* across host development and during transmission

To determine whether non-native *S. capleta* can be maintained during host development, and transmitted to the next generation, egg masses were collected from different *C. alternans* females and divided into two experimental treatments: re-infected with the native symbiont and cross-infected with the symbiont of *C. gressoria*.

Emerged larvae were maintained through pupation and until adulthood. *S. capleta* infection frequency was investigated in the female foregut symbiotic organs and ovary-associated glands 5- and 15 days after adult eclosion, using diagnostic PCR specific to each *S. capleta*-species and as described above. Three to six replicates were performed depending on treatment and sampling time.

Additionally, whole-mount FISH was performed to visualize each *S. capleta* species in the ovary-associated glands, as described above. For each treatment, one female was collected 5 and 15 days after emergence and dissected using sterile scissors. The following oligonucleotide probes were used for in situ hybridization: Atto550-SAL227 targeting the 16S rRNA sequence of *S. capleta* from *C. alternans*, Cy5-SCG224 targeting the 16S rRNA sequence of *S. capleta* from *C. gressoria*, and Atto488-EUKCA164 targeting the 18S rRNA sequence of *C. alternans* (Supplementary Data 5A). Symbiont-targeting probes are specific to each species and were used simultaneously during hybridization.

Three re-infected and two cross-infected females were maintained until sexual maturation. Egg masses laid were recorded daily and subsampled for 1 month to assess their symbiotic status (7 days post oviposition). FISH was applied to paraffin sections to localize symbionts in the egg caplets, as described above. Eggs were dissected and fixed in Carnoy's solution at room temperature overnight. The following oligonucleotide probes were used for in situ hybridization: Atto550-SAL227 targeting the 16S rRNA sequence of *S. capleta* from *C. alternans*, Cy5-SCG224 targeting the 16S rRNA sequence of *S. capleta* from *C. gressoria*, and Atto488-EUKCA164 targeting the 18S rRNA sequence of *C. alternans* (Supplementary Data 5A). Controls consisted of untreated eggs from *C. alternans* and *C. gressoria* beetles (Fig. S7). Two replicates with one-to-four eggs per replicate were carried out for each FISH treatment.

To confirm the FISH results, four egg masses from cross-infected females were tested by diagnostic PCR targeting the non-native *S. capleta*, pooling two egg caplets per egg mass. Egg masses laid by re-infected and cross-infected females were then maintained until larval emergence. The larvae were observed daily and subsampled (3-day-old larvae) to assess their symbiotic status using *S. capleta*-specific diagnostic PCR, as described above. Two and three replicates were performed for cross-infected ($n = 30$ larvae) and re-infected ($n = 48$ larvae) treatments, respectively.

### Dual-infection assays

Dual-infection assays were performed to investigate whether non-native symbionts can colonize *C. alternans* foregut symbiotic organs and propagate in the presence of the native symbiont. Two symbiont-bearing spheres, one bearing the native symbiont and the other collected from a *C. gressoria* egg caplet, were reintroduced into caplet-free eggs. Both spheres were deposited on the anterior pole of each egg and were observed hourly to ensuring that they were both consumed by the embryo before hatching.

To visualize the symbionts in foregut symbiotic organs of dual-infected *C. alternans* larvae, FISH was applied on paraffin sections, as described above. Foregut symbiotic organs from 5-day-old larvae were dissected and fixed in Carnoy's solution at room temperature overnight. The following oligonucleotide probes were used for in situ hybridization: Atto550-SAL227 targeting the 16S rRNA sequence of *S. capleta* from *C. alternans*, Cy5-SCG224 targeting the 16S rRNA sequence of *S. capleta* from *C. gressoria*, and Atto488-EUKCA164 targeting the 18S rRNA sequence of *C. alternans* (Supplementary Data 5A). Symbiont-targeting probes are specific and were used simultaneously during hybridization.

Population dynamics of native and non-native symbionts in foregut symbiotic organs of dual-infected *C. alternans* beetles was estimated through qPCR on an Analytik Jena® qTOWER³ G cycler. Dual-infected larvae were sampled 1-, 5- and 12 days after hatching, and adults (females and males) were sampled 50 days after pupation. Foregut symbiotic organs were dissected, and DNA was extracted using the E.Z.N.A.® Insect DNA Kit from Omega Biotek. To estimate symbiont abundance, the copy number of the *S. capleta groEL* gene from *C. alternans* was quantified with the following primer pair (AGAAGGGATGCAATTTGA-TAGAGGT and GCAACAGATTCTAATATAGGTAAT, Supplementary

Data 5D), and the copy number of the *S. capleta groEL* gene from *C. gressoria* was quantified with the following primer pair (TGAAG-GAATGCAATTTGATAGAGGA and AGCAACAGCTTCTAATATAGGTAAC, Supplementary Data 5D). The primer pairs were designed using Geneious software, and synthesized by Sigma-Aldrich under standard conditions. Primers were designed to be specific to each symbiont but sharing the same amplicon and efficiency. qPCRs were performed using a final reaction volume of 25 μl containing 1 μl of DNA template, 2.5 μl of each primer (10 μM), 6.5 μl of autoclaved distilled H$_2$O, and 12.5 μl of the Platinum SYBR Green qPCR SuperMix-UDG (Thermo Fisher Scientific). The cycling conditions were 3 min of activation at 95 °C, followed by 45 cycles at 95 °C for 15 s, 60 °C for 45 s, 72 °C for 20 s, and a melting curve analysis was conducted by increasing the temperature by 1° every 30 s, from 60 to 95 °C. The gene copy number was estimated from the standard curves (tenfold dilution series from $10^{-1}$ to $10^{-8}$ ng/μl), generated using purified PCR products and measuring their DNA concentration using a NanoDrop™ 1000 spectrophotometer. Treatments were carried out in different experimental blocks. Three to five replicates were performed per treatment.

The relative growth rate of the non-native *S. capleta* compared to the native one in foregut symbiotic organs of *C. alternans* larvae was measured assuming that the mixed infections started at 1:1. The plot of ln(non-native symbiont abundance/ native symbiont abundance) over time was generated with a slope equal to ln$\omega$[105]:

$$\ln\left(\frac{\text{non}-\text{native symbiont abundance(t)}}{\text{native symbiont abundance(t)}}\right) = \ln\left(\frac{\text{non}-\text{native symbiont abundance(0)}}{\text{native symbiont abundance(0)}}\right) + t \cdot \ln(\omega)$$

(1)

The selection coefficient ($\omega$) for non-native bacterium versus native one was calculated using the $e$ raised to the slope of the line fit to the plot of relative *S. capleta* abundance over time[105]. The linear regression to the non-native symbiont growth curve was fitted using R v. 3.5.3[104] and ggplot2[101]. Another method was performed, using the selection rate ($r$) defined as the difference in Malthusian parameters as follows[106]:

$$r = \frac{\left(\ln\left[\frac{\text{density non}-\text{native symbiont at day t}}{\text{density non}-\text{native symbiont at day t}-1}\right] - \ln\left[\frac{\text{density native symbiont at day t}}{\text{density native symbiont at day t}-1}\right]\right)}{\text{day t}}$$

(2)

Egg masses laid by dual-infected adults were maintained until larval emergence. Larvae were observed daily and subsampled (3-day-old larvae) to assess their symbiotic status using *S. capleta*-specific diagnostic PCR, as described above. Three replicates were performed per treatment ($n = 48$ larvae).

## Statistical analyses

To assess the colonization efficiency of different *S. capleta* species in *C. alternans* as a novel host, Fisher's Exact test was used to assess the effect of treatment (untreated, aposymbiotic, re-infected, or cross-infected) on symbiont infection frequency in larval foregut symbiotic organs. Pairwise comparisons between treatments were performed using the pairwise.prop.test() function with Holm correction to adjust for multiple testing. Abundances of different *S. capleta* symbionts in the foregut symbiotic organs of 5-day-old *C. alternans* larvae were analyzed using a generalized linear model with a quasi-Poisson error structure and a log-link function. The treatment, replicate, and experimental block were included in the model as fixed factors. Tukey's HSD pairwise comparisons were performed using the glht() function with Bonferroni corrections. The survival of *C. alternans* beetles into adulthood was analyzed with a Cox mixed-effect model testing for the effect of experimental treatments (untreated, aposymbiotic, re-infected or cross-infected) and replicates with experimental blocks as a random effect, and using the *coxme* package[107]. A pairwise comparison between treatments was performed using the pairwise_survdiff() function with Bonferroni corrections in the

*survminer* package. The survival data were visualized by computing the Kaplan–Meier survival functions[108]. Spearman's rank correlations were used to test the relationship between larval survivorship and symbiont genetic distance (ANI or collinearity score) after testing the data distribution.

To compare the expression level of DEGs after colonization of foregut symbiotic organs by *S. capleta* symbionts, normalized transcripts were analyzed using either a negative binomial generalized linear model implemented by the glm.nb() function, or a general linear model framework depending on the data distribution and after verification that model assumptions were respected. In these statistical models, treatment and replicate were considered as fixed factors. Tukey's HSD pairwise comparisons were performed using the glht() function with Bonferroni corrections. Abundances of different *S. capleta* symbionts in the foregut symbiotic organs of 5-day-old *C. alternans* larvae (RNA-seq experiment) were analyzed using a general linear model after appropriate data transformation and verification of model assumptions. Treatment, replicate, and their interaction were included as fixed factors, and pairwise comparisons were conducted using Tukey's HSD with Bonferroni corrections via the glht() function. Fisher's Exact test was used to assess the effect of treatment (re-infected or cross-infected) on the proportion of *S. capleta* reads mapped to the symbiont genome relative to total reads mapped. Pairwise comparisons between treatments were performed using the pairwise.prop.test() function with Holm correction to adjust for multiple testing. Akaike information criterion was used for model selection.

The effect of experimental treatments (re-infected or cross-infected) on symbiont infection frequencies in foregut symbiotic organs or ovary-associated glands of *C. alternans* females 5- and 15-days post emergence was analyzed using generalized linear models with a binomial error structure and a logit-link function. In these statistical models, treatment, time, and their interaction were considered as fixed factors. The effect of experimental treatments (re-infected or cross-infected adults) on symbiont infection frequency in their offspring was analyzed using a generalized linear model with a binomial error structure and a logit-link function, and the treatment was considered a fixed factor.

To compare the population dynamics of native and non-native bacteria in the foregut symbiotic organs of dual-infected beetles during development, their abundance was analyzed using a general linear model framework after verification that model assumptions were respected. The time, symbiont, replicate, experimental block, and the interaction between the time and symbiont were used as fixed factors. A pairwise comparison of the interaction was performed using least-squares means with Tukey corrections. Given that the data associated with adult foregut symbiotic organs include both females and males, the effect of sex on *S. capleta* population dynamics was determined by fitting a negative binomial generalized linear model implemented by the glm.nb() function. Beetle sex and symbiont were considered as fixed factors. The effect of beetle development time on the abundance of native and non-native symbionts was analyzed using general linear models after checking that model assumptions were respected and using time as a fixed factor. Tukey's HSD pairwise comparisons were performed using the glht() function with Bonferroni corrections. The impact of dual-infected beetles on *S. capleta* infection frequencies in their offspring was analyzed using a generalized linear model with a binomial error structure and a logit-link function, and the treatment was considered a fixed factor. Akaike information criterion was used to select all previous models.

Statistical analyzes were performed in R v. 3.5.3[104] using the prop.test() function to obtain the 95% confidence intervals for a binomial distribution, the *survival* package for survival analyzes[109], *multcomp* and *lsmeans* packages for pairwise comparisons[110,111], *MASS* package for negative binomial generalized linear models[112], and *ggplot2* for producing figures[101].

**Reporting summary**

Further information on research design is available in the Nature Portfolio Reporting Summary linked to this article.

## Data availability

The raw reads of the RNA sequencing data generated in this study have been deposited in the National Center for Biotechnology Information (NCBI) SRA under accession codes SRR31763289-SRR31763297 [https://www.ncbi.nlm.nih.gov/bioproject/?term=PRJNA1199456]. The raw data generated in this study have been deposited in Figshare [https://doi.org/10.6084/m9.figshare.28622969]. Source data are provided with this paper.

## Code availability

The R scripts generated in this study have been deposited in Figshare [https://doi.org/10.6084/m9.figshare.28622969].

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

## Acknowledgements

Financial support from the Max Planck Society, European Research Council (ERC CoG 101171076 "Symbivore" to H.S.), Alexander Von Humboldt Foundation (to I.P.), German Research Foundation (SA 3105/2-1 to H.S.), and European Molecular Biology Organization (to H.S.) is gratefully acknowledged. This study complies with Nagoya Protocol regulations in the Republic of Panama and France per the following permits: ARG-027-2023 and TREL2302365S/693. We thank Brice Menet for his assistance with beetle maintenance and experiments, and Aileen Berasategui for helpful comments on a previous version of the manuscript. We are also grateful to Sven Helfer, Parthena Prasota, and the Microscopy Facility for their technical support, as well as Donald Windsor for sharing his expertise and helping during field trips to collect beetles.

## Author contributions

I.P. and H.S. conceived of the study; I.P., H.S., and R.E.L. designed experiments; I.P., M.G.L., C.E., A.M.A., C.H., and H.E. carried out experiments; I.P. analyzed the data; I.P. and H.S. wrote the manuscript and secured funding. All authors edited and commented on the paper.

## Funding

## Competing interests

The authors declare no competing interests.
