## [Transparent Peer Review file · Nature Communications]

Fidelity in co-diversified symbiosis

Corresponding Author: Dr Hassan Salem

Version 0:

Reviewer comments:

Reviewer #1

(Remarks to the Author)

By employing an experimental symbiont exchange, this study aims to elucidate the evolution of partner recognition and fidelity in symbiotic microorganisms that are strictly vertically transmitted and have reduced genomes. Symbiotic systems with vertical transmission are generally difficult to manipulate experimentally because separating the host and its symbionts is very challenging, making partner-exchange experiments rare. The authors overcame this technical barrier by employing a unique model system, tortoise beetles, wherein symbionts are transmitted via caplets. The primary focus is on a single species of tortoise beetle (*Chelymorpha alternans*), from which the native symbiotic caplet was removed and replaced with caplets from five different tortoise beetle species. This allowed the authors to investigate in detail how co-diversified and genome-reduced symbionts infect, colonize, establish symbiotic associations with different hosts, and are transmitted to the next generation.

Surprisingly (very surprisingly for me), the authors found a high degree of host–symbiont recognition in this vertically transmitted and coevolving symbiotic system. While the native symbiont was well established in the symbiotic organ, conferred substantial fitness benefits to the host, and was reliably transmitted to offspring, non-native symbionts from different species of tortoise beetles were not normally maintained. Although the non-native bacteria initially colonized the symbiotic organ, the host's survivability was reduced, and vertical transmission was unsuccessful. Notably, when non-native symbionts were introduced, immune-related genes were upregulated in the symbiotic organ, suggesting that the host insect discriminates between native and non-native symbionts, even though all have co-diversified and are vertically transmitted. In addition, when native and non-native symbionts were provided simultaneously, the native symbiont gradually became dominant in the symbiotic organ, and only the native symbiont was transmitted to the next generation, despite their initially equal establishment. Through this series of experiments, the authors demonstrate that even in symbiotic systems with strict vertical transmission and co-diversifying, genome-reduced symbionts, host selection and microbial competition are pivotal for maintaining partner fidelity.

As the authors note, similar studies have been conducted in aphids, weevils, and plataspid stinkbugs, but never before has such a precise and extensive experimental framework been applied. In particular, the evidence implicating both microbe–microbe competition and host immune responses as mechanisms underpinning partner fidelity is novel. Furthermore, the results showing that host fitness correlates with the evolutionary distance among symbiotic bacteria (Figure 4C,D), as well as the insights into how hosts respond to coevolving symbionts (Figure 5), are especially noteworthy. These points could be emphasized more explicitly in the abstract and conclusion.

Overall, the manuscript is well written and clearly organized, although there are a few points of concern:

In Figure 4, the authors present experiments in which symbiotic bacteria from different beetle species were introduced into *Chelymorpha alternans*, and all symbionts appeared to colonize the symbiotic organ. In this symbiont exchange experiments, for symbionts from four host species (*Chelymorpha gressoria*, *Acromis sparsa*, *Aspidimorpha quinquefasciata*, and *Cassida rubiginosa*), there was a gradual decline in host survival, and most insects eventually died, similar to the aposymbiotic controls. This result could be due to poor colonization by non-native symbionts, but no quantitative data on symbiont titer in the symbiotic organ are provided. Is the observed mortality a consequence of reduced symbiont density, or is it due to adequate colonization but low activity (e.g., reduced gene expression, as shown in figure 5E)?

The competitive interactions among microorganisms shown in Figure 7 are particularly interesting. One possible mechanism involves the immune response observed in Figure 5. Does simultaneous colonization by native and non-native symbionts also enhance immune activity in the symbiotic organ, thereby eliminating non-native symbionts? (But it seems likely that such immune activation would also impact the native symbiont, too.) Alternatively, does the native symbiont simply outcompete non-native symbionts through superior nutrient acquisition or other fitness advantages within the symbiotic

organ?

Other points:

L69: Spell out *Stammera* in full at first mention.

L92–94: This is a clear summary, but given that multiple symbionts from different lineages were tested, a more detailed discussion of evolutionary distance, fitness effects, and transmission rates (as shown in Figure 4C,D) could be emphasized in this part. This could also be reiterated in the conclusion.

L229: I remain curious about the density of symbiotic bacteria inside the symbiotic organ before this discussion.

Figure 4A: The methods section does not clearly describe how symbiont presence was investigated. It would be helpful to include this information in the text and/or figure legend.

Figure 4: Clearly state the number of biological replicates (n=).

L315–331: Regarding the differential immune responses, it might be informative to inject symbiotic bacteria into the hemolymph and examine immune gene expression dynamics.

L403: It is intriguing that the symbiotic bacteria do not enter the caplet despite colonizing the transmitting organ. Do non-native symbionts attempt to transmit but lose viability within the caplet? Since the authors investigated the caplets only by FISH, the distinction between the hypothesis that symbionts fail to enter the caplet and the hypothesis that they enter but do not survive (less active) is not fully discussed.

(Remarks on code availability)

Raw data and scripts were uploaded appropriately. I don't think there is any problem.

Reviewer #2

(Remarks to the Author)

In this particularly elegant study, Pons et al. use a fascinating example of symbiosis as an experimental system to address a fundamental question in the field: what makes partners in a symbiotic relationship so specific to each other?

First, they demonstrate how tortoise beetle symbioses offer an experimental platform that allows transmission of their respective symbionts to be manipulated and various hypotheses to be tested following reciprocal exchanges. These reciprocal exchanges successfully lead to the infection of their hosts.

They then demonstrate that some of these symbiont exchanges maintain the host's fitness at a level similar to that with its own symbiont, while others lead to a clear decrease in survival rate. The outcomes of the infections therefore differ greatly from one symbiont to another, which correlates with genetic differences between the symbionts.

They then demonstrate that infection with another symbiont can lead to different outcomes in terms of gene expression, even when partially rescuing fitness. They also highlight the potential role of immunity in interactions with non-native symbionts.

Another striking result of the study is that, despite successful infection of larvae and adults, non-native symbionts do not get transmitted to the next generation.

Finally, they demonstrate that, in the context of co-infection with a native symbiont, the non-native symbiont is rapidly outcompeted and lost later on during adulthood.

This is a beautiful and highly original work, offering a striking demonstration of how studying an adequate model system can provide a powerful experimental foundation for addressing a fundamental question in the field, shedding light on the mechanisms behind fidelity in symbiotic associations. I believe this article will contribute significantly to the field of symbiosis and will appeal to a wider audience beyond this (already substantial) area of research.

Overall, I believe the study and its methodology are sound, and the methods are well described. However, I have a few remarks that I would like to be taken into account before publication.

Main remarks

My main concern is the section on how the host recognises the symbiont.

While the results clearly show that the host responds differently to one of the cross-infected symbionts than to its native symbiont, I think the conclusions stretch the immunity hypothesis too far at the expense of other alternative hypotheses. While the data presented here support the immunity hypothesis, they do not fully demonstrate it or rule out other hypotheses, such as fine metabolic interplay between the symbiont and host. While I agree that differential recognition likely occurs, I don't agree that the proposed data demonstrate that this recognition is immunity-based, nor that the resulting effect is immunity-mediated. For example, the assertion in line 344 of a 'heightened immune response' is, in my opinion, overly assertive without providing evidence of increased AMP expression or ROS production, for example.

I think the authors could either conduct more experiments to strengthen the evidence behind this conclusion, or they could be more open in their conclusions, including other hypotheses and developing the potential immunity mechanisms in a follow-up story.

More specifically, I would like to see a more thorough discussion about what NPC2 is. It's also a sterol-binding protein that is involved in many pathways, not just immune-related ones. For instance, it appears to play a part in maintaining intestinal stem cells in *Drosophila*. I think it is too reductive to conclude that the immune pathway is involved based on the change in expression of this gene. To make this claim, I would like to see a measure of the immune response, starting with AMP expression, in the gut of the host in different infection experiments. It would also be interesting to demonstrate whether the decrease in bacterial load is due to the killing of bacteria or the inhibition of their growth. If confirmed, the immune response

itself could also damage the host.

Another possibility is that the fine pairing of the host and symbiont relies on transcriptional crosstalk, which could be necessary for the symbiont to grow properly. This crosstalk could involve NPC2, or it could be independent. Only a loss-of-function experiment (for example, RNA interference against NPC2) could resolve this issue.

The host responds differently to its symbiont and cross-symbiont. However, the link with fitness outcomes could be due to the host not expressing as many digestive enzymes because it is not 'prompted' by the presence of its native symbiont. The transcriptomic data also seem to point towards genes that are more strictly linked to metabolic function than immune function, such as lipase activity. Could the metabolic interplay be disturbed when the symbiont is not the native one?

With regard to cathepsins and serine proteases, while some of them can be involved in immune responses, not all of them are, and they also have pleiotropic functions, including digestion. Their increased expression may also be the host's reaction to reduced nutrient accessibility due to a change in symbionts. Apart from the plant digestive enzymes described by the authors, I would be interested to see whether the comparison between the native symbiont and the symbiont from *A. quinquefasciata* differs in terms of the metabolic advantages they could provide to their hosts, perhaps in relation to lipid metabolism.

In conclusion, I would like to see a more nuanced interpretation of these transcriptomic data.

Finally, while the number of reads is an indication of a decrease in the population of *S. capleta*, a qPCR measurement is required to confirm these results and rule out an effect on the transcriptional activity of the bacteria (which would also be an interesting phenotype).

Minor points

This manuscript is written particularly elegantly, and I enjoyed reading it very much. That said, I have a few comments below that I hope will contribute to improving the manuscript.

The first three sentences of the abstract are somewhat difficult to follow as they encompass many important and complex concepts. I know the abstract is limited in length, but perhaps try rephrasing or simplifying which ones need to be displayed in this short abstract.

Line 16 : why « despite » ?

Lines 22–24: I find it a bit oversimplified to include symbiont recognition and partial rescue in the same sentence, as the link between the two is not fully demonstrated.

Line 30: The first sentence is abrupt; the term 'specialists' could be explained in more detail.

Line 34: Please define or explain what co-diversification is. Also, I'm curious: what is the author's take on the distinction between co-diversification and co-evolution? The term 'co-diversification' is used throughout the manuscript, so it is important to define it clearly.

Line 68 : explain what is meant by « streamlined »

Line 87 : Add 'experimental' before 'mechanism', or reformulate to make it clear that the authors are referring to a way of manipulating their experimental model here.

Line 134-135 : the sentence is not complete

Line 208 : « their effects » : please precise which one(s)

Line 245-248 : The symbionts share the same genes related to plant digestion. Is there any possibility that these genes are not expressed as efficiently in one species as in the other? Apart from the different host response, this could be one hypothesis that the author cannot exclude (at least with the current dataset).

Figure 4B. I wondered why the survival lines stopped even for the insect population that was still alive. I think I now understand that this is because we are only following survival until metamorphosis; these individuals are still alive, but are no longer recorded in the assay. It would be easier for readers if this explanation were included in the figure legend.

Line 469: "Both symbionts share 226 genes but differ in only eight." This is fascinating. I'd love to see a short discussion of what are these 8 genes. Also, is it possible that the genomic difference is very small, but the symbionts regulate differentially their expression? (it is maybe not known yet, but I'm curious to know what the authors think)

Line 481-484: I was a bit lost in this argument that competition increases the specificity of transmission, as the non-native symbiont was shown to not be able to be transmitted to the progeny even when alone. Hence the problem with transmission does not seem to be linked to competition but to an incompatibility between the symbiont and host.

Figure 7A: the native symbiont seems to be closer to the epithelial cells in these FISH examples, compared to the other bacteria. Is it a consistent pattern? If yes can the authors mention it in the text? It could be in link with the symbiont recognition, and/or the out-competitiveness (for example through gaining access to nutrients the out-competed symbiont cannot access because of its regional exclusion).

Conclusion and outlook: I was missing a bit of conclusion on the effect of symbiont replacement on the host fitness, which is also a very important result of this article.

(Remarks on code availability)

Reviewer #3

(Remarks to the Author)

In this work, the authors experimentally test the role of long-term evolutionary codiversification in determining the functional compatibility of mutualistic partners. Using the ancient and specialized symbiosis between *Stammera* bacterial symbionts and their cassidine beetle hosts, they test the hypothesis that symbiotic function (host fitness, microbial fitness, host gene regulation) is associated with the phylogenetic relatedness of the symbiont to the host's native strain. They accomplish this by artificially inoculating eggs with symbiont strains from conspecific and heterospecific beetles, measuring shifts in gene expression and fitness consequences for both host and microbe.

This manuscript was a pleasure to review. The authors designed and undertook experiments that exhibit conceptual elegance and technical virtuosity, which merit the publication of this work in *Nature Communications*. While other research in the field of host-microbe symbiosis have studied coevolution by undertaking similar approaches of subjecting a panel of microbes to a single host, this work is groundbreaking in accomplishing this in a system with a long history of vertical inheritance. In many ways, this is what makes this paper an important and novel contribution to the field. Often coevolutionary interactions and specificity are assumed in these highly co-diversified interactions, but it is often nearly impossible to test these assumptions because host and symbiont are so tightly metabolically integrated. Remarkably, the authors test this important underlying assumption of co-diversified symbiosis by leveraging a naturally occurring interaction. They demonstrate a capacity for creative experimental design, exquisite attention to detail, and commitment to implementing difficult and delicate manipulations in this non-model system. Beyond this, they demonstrate a novel and remarkable feature of this system – the capacity for coinfection. We believe this is a novel contribution for this system, but it is also rare to demonstrate such dynamics broadly in vertically transmitted interactions, especially with the same level of visual spatial resolution as demonstrated here. This paper contributes to a small number of studies showing such coinfection potential. The manuscript was extremely well-written, and the authors do a fantastic job placing this study within the broader context of the field.

The results overall, while not particularly surprising, establish an essential empirical verification of underlying evolutionary assumptions. They not only set the stage for intriguing future experiments but address long-standing questions about the role of coevolution in symbiosis. Perhaps the most striking result is the fact that the co-infected *C. alternans* host still favors the native symbiont over cross-infecting symbionts despite their functional near-equivalency in terms of host fitness and host gene expression, which lends the most credence to their argument that coevolutionary history drives fidelity. The authors could actually emphasize the impact of these results more in their discussion of the results – a symbiont can provide the same level of functional benefit to the host and yet have its transmission inhibited. We would like some discussion of what this means for evolutionary stability and potential. Overall, this is a pretty remarkable result, and this could be brought out more in the discussion.

While we believe this manuscript will make an important contribution, we have some suggestions for improving the overall impact of the findings. First, we believe the discussion of coinfection dynamics could be made stronger if the authors demonstrate the same phenomenon using *C. gressoria* co-infected with the *C. alternans* and native symbionts, ruling out the possibility that the *C. gressoria* symbiont is intrinsically a weaker competitor, perhaps due to differences in metabolic efficiency or localization that cannot be predicted from a comparison of their genomic contents. As is, the coinfection study still offers interesting and important insights, but the reciprocal cross-infection experiment would strengthen these insights. Alternatively, the authors can address this limitation in the text.

Beyond this suggested experiment we also recommend two points of discussion, and one small assay, which could elevate this manuscript, based on our interpretation of the beautiful microscopy that accompanies this work. First, based on Figure 6 and Figure S6 in the supplement, it seems that *C. alternans* OAGs cross-infected with *C. gressoria* *Stammera* exhibit striking morphological remodeling, from a complex labyrinthine architecture to a simpler conglomerate form between 5 and 15 days – it does not seem to occur in *C. alternans* OAGs reinfected with the native *Stammera*. This morphological change is only somewhat clear from the FISH images but is evinced by the *Stammera* signal, which does not shift markedly for the native/magenta *Stammera* but does shift to colocalization with the nuclei(?) for the gressorial/green *Stammera*. Is this an accurate interpretation of these images? Or is this an experimental artifact? The images do appear markedly distinct, so if these morphological differences between treatments at day 15 are not a consequence of a biological effect of symbiont strain, the authors should keep their manuscript as is but could briefly explain the methodological cause of these apparent differences in their response here. If these differences are real, we would appreciate some brief discussion on the potential relevance of this consistent morphological change (as evidenced by Figure S6) for symbiont transmission, and if possible, a conventional bright-field microscopy image that displays at least the surface morphology of CIC vs RI OAGs at 15 days. Ideally, some higher-magnification FISH images of OAGs demonstrating more clearly the spatial shift in the *Stammera* probe signal would be appreciated as well.

Second, based on Figure 7 and Figure S8 in the supplement, it seems that the native symbiont exhibits a distinctive pattern of localization from the gressoria symbiont in the co-colonized foregut organ. The native symbiont seems to monopolize an association with the DAPI-heavy host nuclei as well as being found lining the lumen of the organ; the gressoria symbiont is found exclusively in the lumen. If the authors agree that this is also the case, we suggest discussing the potential implications of differential localization for these symbionts in competition – specifically, whether one might be more prone to displacement over host development than the other. If the authors feel a discussion of strain-specific symbiont localization is informative, I also think it is worth devoting some space in Figure 7 to a panel depicting just an overlay of panels A4 and A5 (or B4 and B5) from Figure S8. The space is there – 7D could be compressed lengthwise, 7E has a lot of white space for just

3 points, and 7F recapitulates the results of a prior experiment.

Minor comments:

Line 63. The authors cite examples of host responses to specialized symbionts that facilitate their selective acquisition. While in these interactions, host do demonstrate striking specificity for recognition of certain symbionts at a broad taxonomic level (i.e. *Vibrio fischeri* or *Caballeronia* spp.), they do not demonstrate fidelity or specificity at a lineage scale. Given these interactions are horizontally transmitted and there is no co-diversification, there is inherently a lack of partner fidelity in the canonical sense (i.e. at a lineage level). As such, we find the phrasing “fidelity... at the genetic and molecular levels” somewhat awkward. We agree that these interactions are maintained by unique molecular and genetics interactions, and that there is fidelity at the species and genus levels respectively, but there are not recorded patterns of fidelity/specificity that typically associated with patterns of local adaptation. Perhaps consider “display striking adaptations for partner recognition at the genetic and molecular levels.”

Line 63 – Reference 37 is a poor fit for the argument made here. While this manuscript does demonstrates the native *Caballeronia* symbiont gains a competitive advantage over non-native symbionts when colonizing *Riptortus* hosts, they do not demonstrate molecular and genetic mechanisms underpinning the competitive success of native *Caballeronia* symbionts. They also do not demonstrate broad patterns of partner specificity by comparing the competitive success between multiple *Caballeronia* strains.

Line 69. This is the first time the name of the symbiont is mentioned outside the abstract, so provide the entire name: *Candidatus Stammera capleta*.

Lines 397-400: If the mechanism for symbiont translocation from the foregut to the OAGs is unknown, the authors could consider weakening this statement to make it clearer that they are speculating the route by which symbiont translocation occurred. The non-native symbiont may use the same unidentified pathway, but it very well may have used an alternative pathway.

Line 811: “considering”- perhaps “assuming”?

Lines 813-815: In its current format this math is cumbersome to interpret. Consider reformatting as an equation.

Across all figures, I strongly urge the inclusion of individual points for the data from each trial on top of all bar plots and box plots using ggplot, to provide the reader with information on experimental consistency within replicate trials for each treatment.

Figure 4: Legend of this figure using inconsistent terminology with the rest of the manuscript and the figure caption. It refers to the untreated control as “caplet intact.” This seems to be the only place in the text where this term is used.

Figure S1ivA, B: These images do not appear to be at the same scale as all the other larval foregut FISH images- provide a revised scale bar.

Figure S5: I am curious about the differential expression of insect OBP A10 in the foregut organs. Do you think this is ectopic expression of an otherwise tissue-specific protein elicited by the symbiont? Or is this just a misannotated transcript?

Figure S6: Is the caption correct? It seems that the top left image corresponds to the merged channels, the top right is host probe, bottom left is gressoria green probe, and bottom right is alternans magenta probe. In addition, the layout of the panels is somewhat confusing- I would be consistent with Day 5 to Day 15 from left to right.

(Remarks on code availability)

We just briefly examined the data files and scripts to ensure they were present and the authors demonstrated transparency. We did not run the scripts ourselves. The R scripts were well annotated and the .csv files appear complete and readable. They did not include a README of the files included themselves, but overall this dataset appears straightforward. These are simply datafiles and scripts for plots and stats. There are no models to review. The data/scripts uploaded is fine as is.

Reviewer #4

(Remarks to the Author)

(Remarks on code availability)

My co-reviewer and I reviewed the data and code together- please see co-reviewer comments.

Version 1:

Reviewer comments:

Reviewer #1

(Remarks to the Author)

The authors have provided precise responses to all my comments, and I am satisfied. The qPCR results are very clear and sufficiently reinforce the results of this study. I have no further comments to add. I really look forward to more in-depth analysis of the mechanisms underlying these symbiotic specificities in the future.

(Remarks on code availability)

The code in this study runs in R and is not particularly special. The codes worked normally.

Reviewer #2

(Remarks to the Author)

The revised version has been largely improved. All the concerns that I had raised have been addressed satisfactorily.

All in one, I think this is a very interesting contribution.

(Remarks on code availability)

Reviewer #3

(Remarks to the Author)

We are pleased with the revisions made to the manuscript and your thoughtful responses to each of our suggestions. Overall, we believe this is interesting work and is ready for publication. We do have one very minor edit. In line 327, you refer to Figure 4B when we think you should refer to Figure 4C. In the text, you refer to the effect of *A. quinquefasciata* on larval rescue, but Figure 4B shows symbiont titer within host. Figure 4C shows the effect of symbiont on survival.

(Remarks on code availability)

We already commented on data availability in the previous review, and we remain pleased.

Reviewer #4

(Remarks to the Author)

(Remarks on code availability)

We assessed the code and data availability previously and we remain satisfied.

REVIEWER COMMENTS

Reviewer #1 (Remarks to the Author):

By employing an experimental symbiont exchange, this study aims to elucidate the evolution of partner recognition and fidelity in symbiotic microorganisms that are strictly vertically transmitted and have reduced genomes. Symbiotic systems with vertical transmission are generally difficult to manipulate experimentally because separating the host and its symbionts is very challenging, making partner-exchange experiments rare. The authors overcame this technical barrier by employing a unique model system, tortoise beetles, wherein symbionts are transmitted via caplets. The primary focus is on a single species of tortoise beetle (*Chelymorpha alternans*), from which the native symbiotic caplet was removed and replaced with caplets from five different tortoise beetle species. This allowed the authors to investigate in detail how co-diversified and genome-reduced symbionts infect, colonize, establish symbiotic associations with different hosts, and are transmitted to the next generation.

Surprisingly (very surprisingly for me), the authors found a high degree of host–symbiont recognition in this vertically transmitted and coevolving symbiotic system. While the native symbiont was well established in the symbiotic organ, conferred substantial fitness benefits to the host, and was reliably transmitted to offspring, non-native symbionts from different species of tortoise beetles were not normally maintained. Although the non-native bacteria initially colonized the symbiotic organ, the host’s survivability was reduced, and vertical transmission was unsuccessful. Notably, when non-native symbionts were introduced, immune-related genes were upregulated in the symbiotic organ, suggesting that the host insect discriminates between native and non-native symbionts, even though all have co-diversified and are vertically transmitted. In addition, when native and non-native symbionts were provided simultaneously, the native symbiont gradually became dominant in the symbiotic organ, and only the native symbiont was transmitted to the next generation, despite their initially equal establishment. Through this series of experiments, the authors demonstrate that even in symbiotic systems with strict vertical transmission and co-diversifying, genome-reduced symbionts, host selection and microbial competition are pivotal for maintaining partner fidelity.

As the authors note, similar studies have been conducted in aphids, weevils, and plataspid stinkbugs, but never before has such a precise and extensive experimental framework been applied. In particular, the evidence implicating both microbe–microbe competition and host immune responses as mechanisms underpinning partner fidelity is novel. Furthermore, the results showing that host fitness correlates with the evolutionary distance among symbiotic bacteria (Figure 4C,D), as well as the insights into how hosts respond to coevolving symbionts (Figure 5), are especially noteworthy. These points could be emphasized more explicitly in the abstract and conclusion.

>>>> We thank the reviewer for the positive assessment and helpful suggestions of our study. We agree that these points could have been emphasized more explicitly in the Abstract and Conclusion sections. We have done so here:

Abstract: ‘We show that non-native *S. capleta* can successfully colonize the symbiotic organs of a novel host, but that the interaction outcome correlates with genetic relatedness to the native symbiont (Lines 19-21).’

Conclusion:

‘We conducted experimental cross-infections in aposymbiotic eggs of the tortoise beetle *C. alternans* using *S. capleta* derived from five additional Cassidinae species. Our results

demonstrate that larval survival is restored following cross-infections, but only when the non-native symbionts are closely genetically related to the native one. Consistent with this, the positive correlation between symbiont genetic distance and host survivorship highlights the strong influence of symbiont genotype on host development (Lines 595-600).'

'The introduction of a genetically distant, non-native symbiont triggered a strong transcriptional response in the host, including activation of immune genes. This was accompanied by a decrease in symbiont population, suggesting host discrimination between native and non-native partners. When both were introduced simultaneously, the native symbiont progressively outcompeted the non-native one and was exclusively transmitted to the next generation (Lines 603-609).'

Overall, the manuscript is well written and clearly organized, although there are a few points of concern:

In Figure 4, the authors present experiments in which symbiotic bacteria from different beetle species were introduced into *Chelymorpha alternans*, and all symbionts appeared to colonize the symbiotic organ. In this symbiont exchange experiments, for symbionts from four host species (*Chelymorpha gressoria*, *Acromis sparsa*, *Aspidimorpha quinquefasciata*, and *Cassida rubiginosa*), there was a gradual decline in host survival, and most insects eventually died, similar to the aposymbiotic controls. This result could be due to poor colonization by non-native symbionts, but no quantitative data on symbiont titer in the symbiotic organ are provided. Is the observed mortality a consequence of reduced symbiont density, or is it due to adequate colonization but low activity (e.g., reduced gene expression, as shown in figure 5E)?

>>>>> We thank the reviewer for this valuable comment. We fully agree that quantitative data on symbiont titer would greatly improve the study and therefore additionally investigated *Stammera* abundance using quantitative PCR. Our results showed that the observed mortality is partly explained by a reduction in symbiont titers, as observed in the cross-infection with the *A. quinquefasciata* symbiont. However, this pattern does not hold for all high-mortality cross-infections, such as those involving the *A. sparsa* symbiont. This suggests that factors beyond symbiont density, such as low metabolic activity, may affect host survival. The manuscript has been revised accordingly:

(Lines 207-220): 'Given the consistent cross-infection of *S. capleta* stemming from six donor beetle species, we further quantified symbiont loads to determine whether infection success was reflected in *S. capleta* abundance. Quantitative analyses revealed treatment-dependent variation in symbiont titer (Figure 4B; GLM (quasi-Poisson), $\chi^2 = 119.32$, d.f. = 7, $p < 0.001$; Table S2B). *S. capleta* abundances were significantly comparable between larvae harboring their native symbiont and those cross-infected with *C. bullata*, *C. gressoria*, and *A. sparsa* symbionts (Figure 4B; pairwise comparison, reinfection: $p = 1$, *C. bullata* symbiont: $p = 1$, *C. gressoria* symbiont: $p = 1$, *A. sparsa* symbiont: $p = 0.084$, Table S2B). Conversely, larvae cross-infected with *A. quinquefasciata* and *C. rubiginosa* symbionts contained lower symbiont titers than the untreated larvae (Figure 4B; pairwise comparison, *A. quinquefasciata*: $p < 0.001$, *C. rubiginosa*: $p < 0.001$, Table S2B). qPCR amplification in aposymbiotic individuals was indistinguishable from background (NTC) levels, indicating that the reported copy numbers are likely overestimations and that aposymbiotic beetles may be devoid of symbionts, as corroborated by the PCR results.'

(Lines 234-241): 'Collectively, this indicates that the observed mortality is only partly explained by a reduction in symbiont abundance, as seen in the cross-infection with the *A. quinquefasciata* symbiont (Figure 4B). However, this pattern does not hold for all high-mortality cross-infections, such as those involving the *A. sparsa* symbiont, where symbiont abundance is comparable to those of the native *S. capleta* (Figure 4B). This variation suggests that host survival following our cross-infections may be influenced by factors beyond symbiont

abundance, including reduced symbiont activity and/or the degree of compatibility between host and microbe. To explore this possibility, we examined whether the mutualistic potential of *S. capleta* in a novel host correlates with its evolutionary relatedness to the native symbiont.' We have revised the manuscript accordingly to reflect these new results.

We have also added the corresponding figure (Figure 4B) and updated the Methods section (Lines 787-805). The revised raw data, statistical tables, and script are included in the supplementary data (Table S2B).

The competitive interactions among microorganisms shown in Figure 7 are particularly interesting. One possible mechanism involves the immune response observed in Figure 5. Does simultaneous colonization by native and non-native symbionts also enhance immune activity in the symbiotic organ, thereby eliminating non-native symbionts? (But it seems likely that such immune activation would also impact the native symbiont, too.) Alternatively, does the native symbiont simply outcompete non-native symbionts through superior nutrient acquisition or other fitness advantages within the symbiotic organ?

>>>> We agree that a potential mechanism underlying the native symbiont's competitive advantage over the non-native symbiont may involve enhanced nutrient acquisition or other fitness-related advantages within the symbiotic organ. To address this point, we have revised the manuscript in line with suggestions from the other reviewers (Lines 558-560): 'The observed distribution may partly explain the native symbiont's competitive advantage, potentially through greater local adaptation and enhanced access to host-derived nutrients.'

Other points:

L69: Spell out *Stammera* in full at first mention.

>>>> Done. We included the entire name, *Candidatus Stammera capleta* (Line 70).

L92–94: This is a clear summary, but given that multiple symbionts from different lineages were tested, a more detailed discussion of evolutionary distance, fitness effects, and transmission rates (as shown in Figure 4C,D) could be emphasized in this part. This could also be reiterated in the conclusion.

>>>> We agree and have now expanded the discussion about the effects of symbiont replacement on host fitness at the end of the introduction (Lines 93-95): 'non-native symbionts can colonize and differentially restore survivorship in a novel host, with outcomes ranging from full fitness recovery to reduced survival depending on genetic divergence among symbionts.'

We also expanded on our findings in the Conclusions section (Lines 595-600): 'We conducted experimental cross-infections in aposymbiotic eggs of the tortoise beetle *C. alternans* using *S. capleta* derived from five additional Cassidinae species. Our results demonstrate that larval survival is restored following cross-infections, but only when the non-native symbionts are closely genetically related to the native one. Consistent with this, the positive correlation between symbiont genetic distance and host survivorship highlights the strong influence of symbiont genotype on host development.'

L229: I remain curious about the density of symbiotic bacteria inside the symbiotic organ before this discussion.

>>>> We were curious as well! Per an earlier response, we have quantified *Stammera* titers across all treatments (Lines 207-220 and 234-241).

Figure 4A: The methods section does not clearly describe how symbiont presence was investigated. It would be helpful to include this information in the text and/or figure legend.

>>>> Thank you for your comment. The sentence has been changed to clearly describe how symbiont presence was investigated (Lines 774-778): 'PCR primers targeting either specific *S. capleta* symbionts (*groEL* gene) or all symbionts collectively (*16S rRNA* gene) were used to verify the symbiotic status of the host beetles (Table S5C). For cross-infected individuals, a *groEL* primer pair specific to the native symbiont was used to confirm its successful removal.'

Figure 4: Clearly state the number of biological replicates (n=).

>>>> Done. We added the number of biological replicates to the Figure 4 legend and elsewhere throughout the manuscript.

L315–331: Regarding the differential immune responses, it might be informative to inject symbiotic bacteria into the hemolymph and examine immune gene expression dynamics.

>>>> We thank the reviewer for this suggestion and agree that such microinjection assays can be extremely informative. We hope they are the basis of a future study, and have revised the manuscript as following (Lines 617-618): 'Moreover, the application of microinjection into the host hemolymph to reproduce the mounted immune response is a worthy area for future study.'

L403: It is intriguing that the symbiotic bacteria do not enter the caplet despite colonizing the transmitting organ. Do non-native symbionts attempt to transmit but lose viability within the caplet? Since the authors investigated the caplets only by FISH, the distinction between the hypothesis that symbionts fail to enter the caplet and the hypothesis that they enter but do not survive (less active) is not fully discussed.

>>>> Thank you for this interesting point. We agree that the distinction is highly informative. Although not elaborated on in our previous submission, we had collected this data as part of our study. Our objective was to assess whether the non-native symbiont can be vertically transmitted to subsequent generations. However, we tested four egg masses from cross-infected females by diagnostic PCR targeting the non-native *S. capleta*, pooling two egg caplets per egg mass. All samples were found to be devoid of symbionts, consistent with the FISH observations and indicating that the non-native symbiont fails to embed within the caplet-associated spheres.

We have added this information (Lines 460-462): 'Diagnostic PCR confirmed the absence of symbionts in eight egg masses tested, demonstrating that the non-native *S. capleta* failed to embed within the maternally secreted spheres.'

And in the Methods section (Lines 889-890): 'To confirm the FISH results, four egg masses from cross-infected females were tested by diagnostic PCR targeting the non-native *S. capleta*, pooling two egg caplets per egg mass.'

Reviewer #1 (Remarks on code availability):

Raw data and scripts were uploaded appropriately. I don't think there is any problem.

>>>> Thank you for your thorough and positive evaluation of our study.

Reviewer #2 (Remarks to the Author):

In this particularly elegant study, Pons et al. use a fascinating example of symbiosis as an experimental system to address a fundamental question in the field: what makes partners in a symbiotic relationship so specific to each other?

First, they demonstrate how tortoise beetle symbioses offer an experimental platform that allows transmission of their respective symbionts to be manipulated and various hypotheses to be tested following reciprocal exchanges. These reciprocal exchanges successfully lead to the infection of their hosts. They then demonstrate that some of these symbiont exchanges maintain the host's fitness at a level similar to that with its own symbiont, while others lead to a clear decrease in survival rate. The outcomes of the infections therefore differ greatly from one symbiont to another, which correlates with genetic differences between the symbionts. They then demonstrate that infection with another symbiont can lead to different outcomes in terms of gene expression, even when partially rescuing fitness. They also highlight the potential role of immunity in interactions with non-native symbionts. Another striking result of the study is that, despite successful infection of larvae and adults, non-native symbionts do not get transmitted to the next generation. Finally, they demonstrate that, in the context of co-infection with a native symbiont, the non-native symbiont is rapidly outcompeted and lost later on during adulthood.

This is a beautiful and highly original work, offering a striking demonstration of how studying an adequate model system can provide a powerful experimental foundation for addressing a fundamental question in the field, shedding light on the mechanisms behind fidelity in symbiotic associations. I believe this article will contribute significantly to the field of symbiosis and will appeal to a wider audience beyond this (already substantial) area of research. Overall, I believe the study and its methodology are sound, and the methods are well described. However, I have a few remarks that I would like to be taken into account before publication.

>>>> Thank you very much for the positive assessment of our study and constructive input which improved the manuscript.

Main remarks

My main concern is the section on how the host recognises the symbiont.

While the results clearly show that the host responds differently to one of the cross-infected symbionts than to its native symbiont, I think the conclusions stretch the immunity hypothesis too far at the expense of other alternative hypotheses. While the data presented here support the immunity hypothesis, they do not fully demonstrate it or rule out other hypotheses, such as fine metabolic interplay between the symbiont and host. While I agree that differential recognition likely occurs, I don't agree that the proposed data demonstrate that this recognition is immunity-based, nor that the resulting effect is immunity-mediated. For example, the assertion in line 344 of a 'heightened immune response' is, in my opinion, overly assertive without providing evidence of increased AMP expression or ROS production, for example.

I think the authors could either conduct more experiments to strengthen the evidence behind this conclusion, or they could be more open in their conclusions, including other hypotheses and developing the potential immunity mechanisms in a follow-up story.

>>>> We agree with the suggestion to adopt a more nuanced approach in our discussion, providing a more balanced interpretation of our transcriptomic data. We elaborated on our changes more specifically below, so please see our response to your queries.

More specifically, I would like to see a more thorough discussion about what NPC2 is. It's also a sterol-binding protein that is involved in many pathways, not just immune-related ones. For

instance, it appears to play a part in maintaining intestinal stem cells in *Drosophila*. I think it is too reductive to conclude that the immune pathway is involved based on the change in expression of this gene. To make this claim, I would like to see a measure of the immune response, starting with AMP expression, in the gut of the host in different infection experiments. It would also be interesting to demonstrate whether the decrease in bacterial load is due to the killing of bacteria or the inhibition of their growth. If confirmed, the immune response itself could also damage the host.

Another possibility is that the fine pairing of the host and symbiont relies on transcriptional crosstalk, which could be necessary for the symbiont to grow properly. This crosstalk could involve NPC2, or it could be independent. Only a loss-of-function experiment (for example, RNA interference against NPC2) could resolve this issue.

The host responds differently to its symbiont and cross-symbiont. However, the link with fitness outcomes could be due to the host not expressing as many digestive enzymes because it is not 'prompted' by the presence of its native symbiont. The transcriptomic data also seem to point towards genes that are more strictly linked to metabolic function than immune function, such as lipase activity. Could the metabolic interplay be disturbed when the symbiont is not the native one?

With regard to cathepsins and serine proteases, while some of them can be involved in immune responses, not all of them are, and they also have pleiotropic functions, including digestion. Their increased expression may also be the host's reaction to reduced nutrient accessibility due to a change in symbionts. Apart from the plant digestive enzymes described by the authors, I would be interested to see whether the comparison between the native symbiont and the symbiont from *A. quinquefasciata* differs in terms of the metabolic advantages they could provide to their hosts, perhaps in relation to lipid metabolism. In conclusion, I would like to see a more nuanced interpretation of these transcriptomic data.

>>>> We thank the reviewer again for the helpful feedback. Per our comment above, we agree and have refined our discussion and developed alternative hypotheses to offer a more balanced approach that extends beyond the immunity angle:

(Lines 346-352): 'Genes encoding Niemann-Pick type C2 (NPC2) proteins were highly expressed in cross-infected larvae colonized by the *A. quinquefasciata* symbiont compared to re-infected control and cross-infected larvae harboring the *C. gressoria* symbiont (Figures 5, C and D; Tables S3, B-E; adjusted $p < 0.05$; *Npc2_1*, NB-GLM, $\chi^2 = 38.7$, d.f. = 2, $p < 0.001$; *Npc2_2*, NB-GLM, $\chi^2 = 35.3$, d.f. = 2, $p < 0.001$; Table S4B). NPC2 proteins are known to be involved in sterol homeostasis, steroid biosynthesis, and in innate immunity by modulating the IMD pathway in insects⁶⁴⁻⁶⁷.'

(Lines 362-366): 'Colonization by the *A. quinquefasciata* symbiont also led to elevated expression of protease-encoding genes, including numerous cysteine cathepsins and a serine protease (Figures 5, C and D; Figure S5; Tables S3, B-E; adjusted $p < 0.05$; NB-GLM/GLM-quasipoisson, $p < 0.001$; Table S4B). These enzymes are known to have pleiotropic roles in digestion, development, and immune function.'

(Lines 373-383): 'Our findings suggest that host immunity may contribute to symbiont recognition. Yet they also raise the possibility that a fine-tuned metabolic interplay between host and symbiont, mediated by transcriptional crosstalk, could be essential for proper symbiont growth. The results further reveal upregulation of genes involved in metabolic functions, such as lipase activity (Figure 5C, Figure S5, Tables S3, B-E; adjusted $p < 0.05$; Table S4B). Such changes could represent a compensatory host response to reduced nutrient access arising from colonization by a metabolically suboptimal symbiont. Consistent with this interpretation, the native (*C. alternans*) and *A. quinquefasciata* symbionts differ in their potential metabolic profiles, especially in lipid metabolism³⁷. *C. alternans* and *C. gressoria*

symbionts encode four and three genes, respectively, related to lipid transport and metabolism, whereas the *A. quinquefasciata* symbiont lacks genes associated with these functions.'

(Line 388): The beginning of the sentence 'Given a heightened immune response ...' has been removed.

(Lines 398-402): 'The pronounced reduction in *A. quinquefasciata* symbiont abundance suggests that an immune response may have inhibited its proliferation within the foregut symbiotic organ of a novel host. However, it remains unclear whether this decrease reflects bacterial killing or reduced growth in an unfavorable host environment.'

(Line 403): 'elevated' has been removed.

(Lines 411-412): 'Surprisingly, non-native *S. capleta* that are closely related to the original symbiont seem to evade immunorecognition by their host.' has been removed.

Moreover, we have revised the manuscript accordingly to reflect these new interpretations (see the Abstract and the Conclusions).

Finally, while the number of reads is an indication of a decrease in the population of *S. capleta*, a qPCR measurement is required to confirm these results and rule out an effect on the transcriptional activity of the bacteria (which would also be an interesting phenotype).

>>>> We thank the reviewer for this valuable comment. We agree and have investigated symbiont abundance using quantitative PCR, and our results confirmed a reduction in the *A. quinquefasciata* symbiont population relative to the native symbiont or the *C. gressoria* symbiont populations. The manuscript has been revised accordingly:

(Lines 384-492): 'We next quantified symbiont abundance using RNA-seq on cDNA and detected significant differences among treatments (Figure 5E; LM, $F_{(2,9)} = 42.12$, $p < 0.001$; Table S4C), consistent with the previously observed proliferation levels of these non-native symbionts (Figure 4B). *S. capleta* titers did not differ significantly between larvae colonized by their native symbiont or cross-infected with the *C. gressoria* symbiont (Figure 5E; pairwise comparison, $p = 1$; Table S4C). In contrast, both groups contained higher symbiont abundance than larvae cross-infected with *S. capleta* from *A. quinquefasciata* (Figure 5E; pairwise comparison, $p < 0.001$ in both cases; Table S4C). We also mapped *S. capleta* reads to measure the relative presence of symbiotic RNA.'

(Lines 398-402): 'The pronounced reduction in *A. quinquefasciata* symbiont abundance suggests that an immune response may have inhibited its proliferation within the foregut symbiotic organ of a novel host. However, it remains unclear whether this decrease reflects bacterial killing or reduced growth in an unfavorable host environment.'

We have also added the corresponding figure (Figure 5E) and updated the methods section (Lines 858-865). The revised raw data, statistical tables (Table S4C), and script are included in the supplementary data.

Minor points

This manuscript is written particularly elegantly, and I enjoyed reading it very much. That said, I have a few comments below that I hope will contribute to improving the manuscript.

The first three sentences of the abstract are somewhat difficult to follow as they encompass many important and complex concepts. I know the abstract is limited in length, but perhaps try rephrasing or simplifying which ones need to be displayed in this short abstract.

>>>> We thank the reviewer and have sought to simplify these three sentences as following (Lines 11-13): 'Obligate co-dependence can arise in symbiosis, yielding heritable partnerships. These interactions are considered to be highly specific, but partner fidelity is difficult to quantify owing to the experimental constraints of symbiont exchange between host species.'

Line 16 : why « despite » ?

>>>> Thank you for the correction. The sentence has been restructured to explain why it is unusual for *Stammera* to possess such a limited metabolism despite existing extracellularly (Lines 16-17): 'Despite its extracellular localization, *S. capleta* possesses a drastically reduced genome (~0.25 Mb) and is vertically transmitted through egg-associated spheres.'

Lines 22–24: I find it a bit oversimplified to include symbiont recognition and partial rescue in the same sentence, as the link between the two is not fully demonstrated.

>>>> We agree and have sought to discuss both findings separately. We have revised both the sentence and the abstract to improve clarity, as follows (Lines 19-26): 'We show that non-native *S. capleta* can successfully colonize the symbiotic organs of a novel host, but that the interaction outcome correlates with genetic relatedness to the native symbiont. Genetically distant species trigger a more pronounced transcriptional response and can only partially rescue host development. While more closely related symbionts proliferate similarly to native one and induce a comparable host response, they fail to propagate to the next generation, underscoring how transmission fidelity, host-symbiont compatibility, and local adaptation can further specificity within a Paleocene-aged partnership.'

Line 30: The first sentence is abrupt; the term 'specialists' could be explained in more detail.

>>>> Done. The sentence has been changed and explained in more detail (Lines 30-31): 'The metabolic intimacy of symbiosis demands the work of specialists, each evolving distinct yet complementary roles to sustain the partnership.'

Line 34: Please define or explain what co-diversification is. Also, I'm curious: what is the author's take on the distinction between co-diversification and co-evolution? The term 'co-diversification' is used throughout the manuscript, so it is important to define it clearly.

>>>> We completely agree that defining 'co-diversification' is key to many aspects of our manuscript and have done so in lines 34–35: 'As symbiont and host co-diversify, they exhibit parallel lineage splitting over evolutionary timescales.'

Line 68 : explain what is meant by « streamlined »

>>>> We meant to convey that the symbiosis is predicated on few host-beneficial factors. However, we see how the word may not be immediately helpful and have removed from the text (Line 68).

Line 87 : Add 'experimental' before 'mechanism', or reformulate to make it clear that the authors are referring to a way of manipulating their experimental model here.

>>>>> We agree and have added 'experimental' before 'mechanism' (Line 88).

Line 134-135 : the sentence is not complete

>>>>> Thank you, the sentence has been changed (Lines 136-138): 'On the other hand, cross-infected embryos were successfully colonized by the symbiont of *C. alternans* (Figure 2B; Figure S1ii), indicating that the cross-infection protocol was reciprocal.'

Line 208 : « their effects » : please precise which one(s)

>>>>> We agree and have changed the sentence as following (Lines 221-223): 'We next quantified how these non-native symbionts shaped the development of *C. alternans* larvae and found that their effects on host survival varied, yielding a gradient of mutualistic outcomes in larvae.'

Line 245-248 : The symbionts share the same genes related to plant digestion. Is there any possibility that these genes are not expressed as efficiently in one species as in the other? Apart from the different host response, this could be one hypothesis that the author cannot exclude (at least with the current dataset).

>>>>> We thank the reviewer for this comment. We agree, we have added this hypothesis to the revised manuscript (Lines 269–271): 'It is conceivable that differences in the expression or regulation of these host-beneficial factors could contribute to the observed variation.'

Figure 4B. I wondered why the survival lines stopped even for the insect population that was still alive. I think I now understand that this is because we are only following survival until metamorphosis; these individuals are still alive, but are no longer recorded in the assay. It would be easier for readers if this explanation were included in the figure legend.

>>>>> Absolutely. Thank you for this helpful comment. We agree that additional clarification was required, and we have now added the following explanation to the Figure 4 legend: 'Beetle survival was monitored until metamorphosis; truncated lines denote surviving individuals censored from the assay'.

Line 469: "Both symbionts share 226 genes but differ in only eight." This is fascinating. I'd love to see a short discussion of what are these 8 genes. Also, is it possible that the genomic difference is very small, but the symbionts regulate differentially their expression? (it is maybe not known yet, but I'm curious to know what the authors think)

>>>>> Thank you very much for this valuable comment. We now included this information in our revised manuscript. As shown in Table S9, the *C. alternans* symbiont uniquely possesses genes associated with translation and energy metabolism (*serS*, *gapA*, *fldA*), suggesting roles in protein synthesis and energy production. In contrast, the *C. gressoria* symbiont encodes genes involved in amino acid and cell wall biosynthesis (*glyA*, *murA*, *rlmE*), indicating contributions to structural maintenance and metabolic support. Although only eight genes differ between the two symbionts, the competitive advantage of the *C. alternans* symbiont may be attributed to these specific genetic differences and/or to variations in gene expression regulation that lead to distinct physiological capacities. However, since we did not generate a multi-RNA-seq dataset for this study, this remains a hypothesis.

We have added this discussion to the revised manuscript (Lines 530-535): 'The *C. alternans* symbiont additionally possesses genes associated with translation and energy metabolism,

whereas the *C. gressoria* symbiont encodes genes involved in amino acid and cell wall biosynthesis, reflecting differences in metabolic specialization (Figure S9). The variations in symbiont population growth may be attributed to these specific genetic differences and/or to variations in the regulation of gene expression that led to distinct physiological capacities.'

Line 481-484: I was a bit lost in this argument that competition increases the specificity of transmission, as the non-native symbiont was shown to not be able to be transmitted to the progeny even when alone. Hence the problem with transmission does not seem to be linked to competition but to an incompatibility between the symbiont and host.

>>>> Thank you. We have removed the sentence and revised the discussion accordingly, consistent with this and other reviewers' comments (Lines 551-557): 'Our findings show that once both symbionts colonize the foregut symbiotic organs, the native population rapidly supplants the non-native one, despite their near-functional equivalence and effects on host fitness (Figure 4) and metabolic response (Figure 5). This indicates that the native bacterium is better adapted to its host and reinforces the hypothesis that co-evolutionary history is a key driver of fidelity.'

Figure 7A: the native symbiont seems to be closer to the epithelial cells in these FISH examples, compared to the other bacteria. Is it a consistent pattern? If yes can the authors mention it in the text? It could be in link with the symbiont recognition, and/or the out-competitiveness (for example through gaining access to nutrients the out-competed symbiont cannot access because of its regional exclusion).

>>>> We thank the reviewer for this important observation. Spatial heterogeneity between both symbiont populations indeed seems to occur, with the native symbiont occupying the area adjacent to the epithelium, whereas the *C. gressoria* symbiont is localized exclusively in the lumen.

We have clarified this point in the revised manuscript and discussed its potential implications for symbiont competition (Lines 555-560): 'We further observed spatial heterogeneity in symbiont distribution across the symbiotic organs. The native bacterium predominantly occupies regions adjacent to the epithelium, whereas the non-native bacterium remains restricted to the lumen (Figures 7, Bi-iv; Figure S8). The observed distribution may partly explain the native symbiont's competitive advantage, potentially through greater local adaptation and enhanced access to host-derived nutrients.'

Conclusion and outlook: I was missing a bit of conclusion on the effect of symbiont replacement on the host fitness, which is also a very important result of this article.

>>>> Thank you for this comment. We have expanded the discussion about the effects of symbiont replacement on the host fitness (Lines 595-600): 'We conducted experimental cross-infections in aposymbiotic eggs of the tortoise beetle *C. alternans* using *S. capleta* derived from five additional Cassidinae species. Our results demonstrate that larval survival is restored following cross-infections, but only when the non-native symbionts are closely genetically related to the native one. Consistent with this, the positive correlation between symbiont genetic distance and host survivorship highlights the strong influence of symbiont genotype on host development.'

Reviewer #3 (Remarks to the Author):

In this work, the authors experimentally test the role of long-term evolutionary codiversification in determining the functional compatibility of mutualistic partners. Using the ancient and specialized symbiosis between *Stammera* bacterial symbionts and their cassidine beetle hosts, they test the hypothesis that symbiotic function (host fitness, microbial fitness, host gene regulation) is associated with the phylogenetic relatedness of the symbiont to the host's native strain. They accomplish this by artificially inoculating eggs with symbiont strains from conspecific and heterospecific beetles, measuring shifts in gene expression and fitness consequences for both host and microbe.

This manuscript was a pleasure to review. The authors designed and undertook experiments that exhibit conceptual elegance and technical virtuosity, which merit the publication of this work in *Nature Communications*. While other research in the field of host-microbe symbiosis have studied coevolution by undertaking similar approaches of subjecting a panel of microbes to a single host, this work is groundbreaking in accomplishing this in a system with a long history of vertical inheritance. In many ways, this is what makes this paper an important and novel contribution to the field. Often coevolutionary interactions and specificity are assumed in these highly co-diversified interactions, but it is often nearly impossible to test these assumptions because host and symbiont are so tightly metabolically integrated. Remarkably, the authors test this important underlying assumption of co-diversified symbiosis by leveraging a naturally occurring interaction. They demonstrate a capacity for creative experimental design, exquisite attention to detail, and commitment to implementing difficult and delicate manipulations in this non-model system. Beyond this, they demonstrate a novel and remarkable feature of this system – the capacity for coinfection. We believe this is a novel contribution for this system, but it is also rare to demonstrate such dynamics broadly in vertically transmitted interactions, especially with the same level of visual spatial resolution as demonstrated here. This paper contributes to a small number of studies showing such coinfection potential. The manuscript was extremely well-written, and the authors do a fantastic job placing this study within the broader context of the field.

>>>> Thank you very much for the positive assessment and constructive input throughout.

The results overall, while not particularly surprising, establish an essential empirical verification of underlying evolutionary assumptions. They not only set the stage for intriguing future experiments but address long-standing questions about the role of coevolution in symbiosis. Perhaps the most striking result is the fact that the co-infected *C. alternans* host still favors the native symbiont over cross-infecting symbionts despite their functional near-equivalency in terms of host fitness and host gene expression, which lends the most credence to their argument that coevolutionary history drives fidelity. The authors could actually emphasize the impact of these results more in their discussion of the results – a symbiont can provide the same level of functional benefit to the host and yet have its transmission inhibited. We would like some discussion of what this means for evolutionary stability and potential. Overall, this is a pretty remarkable result, and this could be brought out more in the discussion.

>>>> We thank the reviewers for the constructive feedback. We have emphasized the impact of the co-infection experiment from an evolutionary stability perspective in our discussion (Lines 555-559): 'Our findings show that once both symbionts colonize the foregut symbiotic organs, the native population rapidly supplants the non-native one, despite their near-functional equivalence and effects on host fitness (Figure 4) and metabolic response (Figure 5). This indicates that the native bacterium is better adapted to its host and reinforces the hypothesis that co-evolutionary history is a key driver of fidelity.'

While we believe this manuscript will make an important contribution, we have some suggestions for improving the overall impact of the findings. First, we believe the discussion

of coinfection dynamics could be made stronger if the authors demonstrate the same phenomenon using *C. gressoria* co-infected with the *C. alternans* and native symbionts, ruling out the possibility that the *C. gressoria* symbiont is intrinsically a weaker competitor, perhaps due to differences in metabolic efficiency or localization that cannot be predicted from a comparison of their genomic contents. As is, the coinfection study still offers interesting and important insights, but the reciprocal cross-infection experiment would strengthen these insights. Alternatively, the authors can address this limitation in the text.

>>>> We thank the reviewers for this valuable comment. We agree that a reciprocal cross-infection experiment would further strengthen our conclusions and have thus addressed this limitation in the revised manuscript (Lines 560-562): 'We note that a reciprocal dual-infection experiment in *C. gressoria* would help rule out the alternative possibility that the *C. gressoria* symbiont is intrinsically a weaker competitor due to differences in its metabolic relative to *S. capleta* associated with *C. alternans*.'

Beyond this suggested experiment we also recommend two points of discussion, and one small assay, which could elevate this manuscript, based on our interpretation of the beautiful microscopy that accompanies this work. First, based on Figure 6 and Figure S6 in the supplement, it seems that *C. alternans* OAGs cross-infected with *C. gressoria* Stammera exhibit striking morphological remodeling, from a complex labyrinthine architecture to a simpler conglomerate form between 5 and 15 days- it does not seem to occur in *C. alternans* OAGs reinfected with the native Stammera. This morphological change is only somewhat clear from the FISH images but is evinced by the Stammera signal, which does not shift markedly for the native/magenta Stammera but does shift to colocalization with the nuclei(?) for the gressorial/green Stammera. Is this an accurate interpretation of these images? Or is this an experimental artifact? The images do appear markedly distinct, so if these morphological differences between treatments at day 15 are not a consequence of a biological effect of symbiont strain, the authors should keep their manuscript as is but could briefly explain the methodological cause of these apparent differences in their response here. If these differences are real, we would appreciate some brief discussion on the potential relevance of this consistent morphological change (as evidenced by Figure S6) for symbiont transmission, and if possible, a conventional bright-field microscopy image that displays at least the surface morphology of CIC vs RI OAGs at 15 days. Ideally, some higher-magnification FISH images of OAGs demonstrating more clearly the spatial shift in the Stammera probe signal would be appreciated as well.

>>>> We thank the reviewers for raising this point and agree that the ovary-associated glands cross-infected with the *C. gressoria* symbiont seem to undergo marked morphological remodeling, shifting from a complex labyrinthine architecture to a simpler conglomerate form between 5 and 15 days.

We have clarified this point in the revised manuscript and discussed its potential implications for symbiont transmission (Lines 464-4469): 'The ovary-associated glands cross-infected with the *C. gressoria* symbiont appear to undergo marked morphological remodeling compared to those infected by the native symbiont, from a complex labyrinthine architecture to a simpler conglomerate form between 5 and 15 days (Figures 6D; Figure S6). How, and whether, such a morphological change affects to the failure to transmit the non-native symbiont is unknown.'

Second, based on Figure 7 and Figure S8 in the supplement, it seems that the native symbiont exhibits a distinctive pattern of localization from the *gressoria* symbiont in the co-colonized foregut organ. The native symbiont seems to monopolize an association with the DAPI-heavy host nuclei as well as being found lining the lumen of the organ; the *gressoria* symbiont is found exclusively in the lumen. If the authors agree that this is also the case, we suggest

discussing the potential implications of differential localization for these symbionts in competition- specifically, whether one might be more prone to displacement over host development than the other. If the authors feel a discussion of strain-specific symbiont localization is informative, I also think it is worth devoting some space in Figure 7 to a panel depicting just an overlay of panels A4 and A5 (or B4 and B5) from Figure S8. The space is there- 7D could be compressed lengthwise, 7E has a lot of white space for just 3 points, and 7F recapitulates the results of a prior experiment.

>>>> We thank the reviewers for this insightful observation. We completely agree. Spatial heterogeneity indeed seems to occur, with the native symbiont occupying the area adjacent to the epithelium, whereas the *C. gressoria* symbiont is localized exclusively in the lumen. Reviewer 2 similarly commented on this pattern.

We have clarified this point in the revised manuscript and discussed its potential implications for symbiont competition (Lines 555-560): 'We further observed spatial heterogeneity in symbiont distribution across the symbiotic organs. The native bacterium predominantly occupies regions adjacent to the epithelium, whereas the non-native bacterium remains restricted to the lumen (Figures 7, Bi-iv; Figure S8). The observed distribution may partly explain the native symbiont's competitive advantage, potentially through greater local adaptation and enhanced access to host-derived nutrients.'

In addition, we changed Figure 7 by adding panels B4 and B5 from Figure S8.

Minor comments:

Line 63. The authors cite examples of host responses to specialized symbionts that facilitate their selective acquisition. While in these interactions, host do demonstrate striking specificity for recognition of certain symbionts at a broad taxonomic level (i.e. *Vibrio fischeri* or *Caballeronia* spp.), they do not demonstrate fidelity or specificity at a lineage scale. Given these interactions are horizontally transmitted and there is no co-diversification, there is inherently a lack of partner fidelity in the canonical sense (i.e. at a lineage level). As such, we find the phrasing "fidelity... at the genetic and molecular levels" somewhat awkward. We agree that these interactions are maintained by unique molecular and genetics interactions, and that there is fidelity at the species and genus levels respectively, but there are not recorded patterns of fidelity/specificity that typically associated with patterns of local adaptation. Perhaps consider "display striking adaptations for partner recognition at the genetic and molecular levels."

>>>> We agree and have changed the sentence as requested (Line 64).

Line 63 – Reference 37 is a poor fit for the argument made here. While this manuscript does demonstrates the native *Caballeronia* symbiont gains a competitive advantage over non-native symbionts when colonizing *Riptortus* hosts, they do not demonstrate molecular and genetic mechanisms underpinning the competitive success of native *Caballeronia* symbionts. They also do not demonstrate broad patterns of partner specificity by comparing the competitive success between multiple *Caballeronia* strains.

>>>> Thank you for the comment. We have removed the reference from that sentence.

Line 69. This is the first time the name of the symbiont is mentioned outside the abstract, so provide the entire name: *Candidatus Stammera capleta*.

>>>> Thank you. We wrote the entire name, *Candidatus Stammera capleta* (Line 70).

Lines 397-400: If the mechanism for symbiont translocation from the foregut to the OAGs is unknown, the authors could consider weakening this statement to make it clearer that they are speculating the route by which symbiont translocation occurred. The non-native symbiont may use the same unidentified pathway, but it very well may have used an alternative pathway.

>>>> We agree with your comment. The statement has been modified to clarify that this is speculation (Lines 451-454): 'The mechanism by which symbionts translocate from the foregut symbiotic organs to the ovary-associated glands is still unknown, but it is conceivable that the non-native symbiont can coopt this pathway to reach the transmission organs of its novel host (Figures 6, B-D; Figure S6).'

Line 811: "considering"- perhaps "assuming"?

>>>> Done. We replaced 'considering' with 'assuming' (Line 933).

Lines 813-815: In its current format this math is cumbersome to interpret. Consider reformatting as an equation.

>>>> We agree. The math has been reformatted as equations (Line 937 and Line 945).

Across all figures, I strongly urge the inclusion of individual points for the data from each trial on top of all bar plots and box plots using ggplot, to provide the reader with information on experimental consistency within replicate trials for each treatment.

>>>> We agree. Dots indicating replicates or the average of each replicate have been added to each bar plot and box plot. The figure legends and scripts have been modified accordingly.

Figure 4: Legend of this figure using inconsistent terminology with the rest of the manuscript and the figure caption. It refers to the untreated control as "caplet intact." This seems to be the only place in the text where this term is used.

>>>> We agree. The terminology in Figure 4 and its legend has been changed to be consistent with the rest of the manuscript.

Figure S1ivA, B: These images do not appear to be at the same scale as all the other larval foregut FISH images- provide a revised scale bar.

>>>> Thank you. These images have been modified to be on the same scale as all other larval foregut FISH images.

Figure S5: I am curious about the differential expression of insect OBP A10 in the foregut organs. Do you think this is ectopic expression of an otherwise tissue-specific protein elicited by the symbiont? Or is this just a misannotated transcript?

>>>> Excellent point. We are confident that this is not a misannotated transcript. In addition to our manual annotation of this gene, OBP has been previously associated with symbiont survival and vertical transmission in plataspid stinkbugs (Koga et al. 2021, PNAS) and with hemocyte regulation by the obligate mutualist *Wigglesworthia* in tsetse flies (Benoit et al. 2017, eLife). These findings suggest that this gene may play a role in mediating the specificity of the *S. capleta* – *Cassidinae* symbiosis.

Figure S6: Is the caption correct? It seems that the top left image corresponds to the merged channels, the top right is host probe, bottom left is gressoria green probe, and bottom right is

alternans magenta probe. In addition, the layout of the panels is somewhat confusing- I would be consistent with Day 5 to Day 15 from left to right.

>>>> Thank you very much. The legend has been corrected and the layout of the panels redesigned.

Reviewer #3 (Remarks on code availability):

We just briefly examined the data files and scripts to ensure they were present and the authors demonstrated transparency. We did not run the scripts ourselves. The R scripts were well annotated and the .csv files appear complete and readable. They did not include a README of the files included themselves, but overall this dataset appears straightforward. These are simply datafiles and scripts for plots and stats. There are no models to review. The data/scripts uploaded is fine as is.

>>>> Thank you for your thorough and positive evaluation of our study.

Reviewer #4 (Remarks to the Author):

>>>> Thank you for helping us improve our study.

REVIEWERS' COMMENTS

Reviewer #1 (Remarks to the Author):

The authors have provided precise responses to all my comments, and I am satisfied. The qPCR results are very clear and sufficiently reinforce the results of this study. I have no further comments to add. I really look forward to more in-depth analysis of the mechanisms underlying these symbiotic specificities in the future.

Reviewer #1 (Remarks on code availability):

The code in this study runs in R and is not particularly special. The codes worked normally.

>>>>> We thank you for your positive feedback on our study.

Reviewer #2 (Remarks to the Author):

The revised version has been largely improved. All the concerns that I had raised have been addressed satisfactorily.

All in one, I think this is a very interesting contribution.

>>>>> Thank you very much for your favorable evaluation of our work.

Reviewer #3 (Remarks to the Author):

We are pleased with the revisions made to the manuscript and your thoughtful responses to each of our suggestions. Overall, we believe this is interesting work and is ready for publication. We do have one very minor edit. In line 327, you refer to Figure 4B when we think you should refer to Figure 4C. In the text, you refer to the effect of *A. quinquefasciata* on larval rescue, but Figure 4B shows symbiont titer within host. Figure 4C shows the effect of symbiont on survival.

Reviewer #3 (Remarks on code availability):

We already commented on data availability in the previous review, and we remain pleased.

>>>>> Thank you very much for this positive assessment and for pointing out this error. The reference to Figure 4B has now been updated to Figure 4C.

Reviewer #4 (Remarks to the Author):

Reviewer #4 (Remarks on code availability):

We assessed the code and data availability previously and we remain satisfied.

>>>>> We thank you for your favorable evaluation of our study.